# Enhancing surface activity and durability in triple conducting electrode for protonic ceramic electrochemical cells

Shuanglin Zheng [1], Wei Wu[2], Yuchen Zhang[2], Zeyu Zhao [2], Chuancheng Duan[3], Saroj Karki[1] & Hanping Ding [1]

With the material system operating at lower temperatures, protonic ceramic electrochemical cells (PCECs) can offer high energy efficiency and reliable performance for both power generation and hydrogen production, making them a promising technology for reversible energy cycling. However, PCEC faces technical challenges, particularly regarding electrode activity and durability under high current density operations. To address these challenges, we introduce a nano-architecture oxygen electrode characterized by high porosity and triple conductivity, designed to enhance catalytic activity and interfacial stability through a self-assembly approach, while maintaining scalability. Electrochemical cells incorporating this advanced electrode demonstrate robust performance, achieving a peak power density of 1.50 W cm$^{-2}$ at 600 °C in fuel cell mode and a current density of 5.04 A cm$^{-2}$ at 1.60 V in electrolysis mode, with enhanced stability on transient operations and thermal cycles. The underlying mechanisms are closely related to the improved surface activity and mass transfer due to the dual features of the electrode structure. Additionally, the enhanced interfacial bonding between the oxygen electrode and electrolyte contributes to increased durability and thermomechanical integrity. This study underscores the critical importance of optimizing electrode microstructure to achieve a balance between surface activity and durability.

High-temperature solid oxide cells (SOCs) are a promising technology for reversible fuel cell and electrolysis applications due to their high energy efficiency. They can store excess renewable energy as hydrogen and generate electricity when needed. Recent advancements in the materials, structures, and manufacturing of conventional oxygen-ion conducting SOCs have significantly enhanced their performance and long-term stability, potentially reducing costs[1-3]. However, the high operating temperatures required still pose challenges, including rapid degradation, high stack costs, and thermal incompatibility. Therefore, lowering the operating temperature is crucial to

addressing these issues[4-6]. Protonic ceramic electrochemical cells offer a compelling alternative, demonstrating improved reliability, energy efficiency, and cost-effectiveness for hydrogen production or electricity generation, making them suitable for lower-temperature operations (350 ~ 500 °C). These benefits arise from the high proton conductivity of their electrolyte, better material compatibility, and favorable electrode reaction kinetics[7,8].

Nevertheless, the reduction of PCEC operation temperature accordingly requires the matching of fast reaction activity on oxygen electrodes to result in the low electrode polarization resistance, since

[1]School of Aerospace and Mechanical Engineering, University of Oklahoma, Norman, OK 73019, USA. [2]Energy and Environment Science & Technology, Idaho National Laboratory, Idaho Falls, ID 83415, USA. [3]Department of Chemical Engineering, University of Utah, Salt Lake City, UT 84112, USA. e-mail: hding@ou.edu

it is crucial for facilitating the electrochemical reactions, in particular, under high current density and steam concentration. Thus, the proper electrode is needed to adapt into the electrochemical material system that is consisted of proton conducting electrolyte, which typically demands efficient transport and dissociation of reaction species, stability, thermochemical compatibility with electrolyte[9–11]. To be considered as the rational composition, the triple phase conductivity of electron, proton and oxygen ion is needed to extend the reaction sites throughout the entire electrode layer for maximizing catalytic activity[12,13]. Recent progresses have been made on the development of several promising triple conducting oxides such as $BaCo_{0.4}Fe_{0.4}Zr_{0.4}Y_{0.1}O_{3-\delta}$ (BCFZY), $PrBa_{0.5}Sr_{0.5}Co_{1.5}Fe_{0.5}O_{5+\delta}$ (PBSCF), $Pr_2NiO_{4+\delta}$ (PNO), and $PrNi_{0.7}Co_{0.3}O_{3-\delta}$ (PNC) electrodes which demonstrate superior fuel cell or electrolysis performances at reduced temperature range (350–500 °C)[14–17]. These electrodes undoubtedly are proved to possess a significant fraction of proton conduction that is triggered by the formation of proton defects in the lattice by different mechanisms. The underlying mechanisms have been investigated to elucidate the intrinsic factors that support the favorable electronic structure for proton defect formation and transport. Additionally, to further enhance PCEC electrochemical activity, recently there are several other approaches being adopted to enhance the reaction kinetics at the interface where ions, electrons and gas meet[18,19].

The durable operation of PCECs depends on several critical factors: rapid kinetic reactions, effective charge and mass transfer, and reliable interfacial bonding. These factors are dynamically linked and influence long-term degradation under complex operating conditions. To meet these requirements, precise control over electrode composition, morphology, layer microstructure, and interface is essential[20–22]. This ensures optimal surface activity, ion diffusion, and mass transfer while maintaining the cell structure and local geometry without deterioration. The electrode particle structure plays a key role, with micron-sized particles stacked to create porosity and active surfaces for reactions. Traditionally, electrodes in PCECs are made by sintering regularly shaped particles, which are randomly stacked to form the electrode layer. This structure relies on the contact between solid phases and the open spaces between them, which may not be optimal[23–25]. Additionally, the interfacial bonding between electrode particles and the electrolyte membrane after thermal treatment is limited by the contact area, affecting long-term stability and interfacial polarization resistance[26,27].

To address these challenges, we have developed a nanofiber-architecture ultra-porous (NAUP) electrode, which integrates ultrafine nanoparticles and hollow gas pathways to significantly enhance surface activity, mass transfer, and interfacial stability. This novel design leverages the unique properties of a triple-conducting composition, enabling the simultaneous conduction of protons, oxygen ions, and electrons. Such triple-conducting behavior facilitates seamless electron and ion diffusion while supporting rapid reaction kinetics and bulk diffusion processes, creating a highly synergistic system. At the macroscopic level, the NAUP electrode's channel-structured fibers provide high porosity, enabling efficient mass transfer of steam to supply adequate reactants for hydrogen production. Concurrently, the nano-scale electrode particles offer an enhanced surface area for catalytic activity, accelerating reaction kinetics. This synergy forms a feedback loop in which efficient mass transfer supports rapid reaction kinetics, and in turn, the accelerated reactions further drive mass transport. At the microscopic level, the unique triple-conducting nature of the PNC material ensures efficient surface reactions and bulk transport. Protons are swiftly conducted through the electrode, while oxygen ion and electron conduction facilitate the removal of intermediates, reducing the accumulation of reaction by-products and further enhancing reaction efficiency.

The synergistic integration of these structural and functional advancements translates into improved electrochemical performance. The NAUP electrode exhibits reduced activation energy for ORR, as well as minimized polarization resistance, leading to enhanced current density and cost-effectiveness. Under practical conditions, the NAUP electrode achieved a peak power density of 1.50 W cm$^{-2}$ at 600 °C in fuel cell mode and a current density of 5.04 A cm$^{-2}$ at 1.60 V in electrolysis mode, with the potential to achieve faradaic efficiency greater than 90% at 1.0 A cm$^{-2}$, which proves the excellence in recent PCEC progresses (Supplementary Fig. 1, Supplementary Table 1 and Supplementary Table 2). The substantial improvement in electrochemical activity is attributed to the reduced electrode polarization resistance, which results from enhanced surface activity, improved gas transport, and optimized interfacial morphology. The durability of the NAUP electrode under transient reversible operation, thermal cycling, and various challenging conditions has been demonstrated to be robust and reliable for practical applications. The readily manufactured electrode and its scalability will potentially accelerate its adoption into PCEC systems.

## Results

### Formation of self-assembled NAUP electrode structure

To facilitate the spontaneous self-assembled synthesis of the electrode component, we used a natural cotton mesh as a template, which provided the necessary mechanical strength. As illustrated in Fig. 1a, after being properly tailored, the mesh was entirely immersed into the triple conducting $PrNi_{0.7}Co_{0.3}O_{3-\delta}$ (PNC73) alcoholic solution for 2 hours for absorbing sufficient loading with stoichiometric composition.Upon soaking, the color of the saturated mesh changed from white to dark green. To preserve its flatness during the subsequent calcination process, the soaked mesh was carefully placed between two aluminum plates for sintering at 750 °C for 3 hours to form black NAUP textile electrode, which was consequently cut to a precise area of 0.178 cm² as the functional electrode layer. In the conventional manufacturing process for the powder-packed PNC73 electrode, a thoroughly mixed PNC73 ink combined with a binder and disperser ensures strong bonding between interfaces. Accordingly, 10 µl of PNC73 ink was also applied to adhere the mesh-structured electrode to the surface of the electrolyte.

Once the PNC73 ink has been applied to the mesh-structured electrode using a pipette, the full cells are dried in the oven for 1 hour. Following this, a second calcination process is carried out at 1000 °C for 2 hours. The morphology of the mesh-structured electrode was examined using a scanning electron microscope (SEM), which clearly displayed the interlaced textile mesh fibers in images of meshes soaked in both 1 M and 2 M solutions. This confirms the porous properties and suitable mechanical flexibility of the structure (Fig. 1b, Supplementary Fig. 2 and 3a).

By subjecting the PNC73-loaded mesh to sintering at temperatures well above the melting point of cotton fabric, the material was successfully preserved in its original self-architecture mesh shape. The moderate porosity level, around 50%, of the oxygen electrode has been shown to exhibit optimal electrochemical performance[25]. Therefore, the mesh-structured electrode immersed in 1 M PNC73 solution was used for all full cells in this study.

To further clarify the uniformity of the mesh-structured electrode, energy-dispersive X-ray (EDS) mapping images are shown in Fig. 1c. These images reveal the presence of elements Pr, Ni, Co, and O distributed consistently across all mesh fiber bundles, indicating a uniform distribution of the desired elements in the processed electrode samples. Figure 1d demonstrates the optimal porosity of the electrode, with a SEM image of a single mesh fiber in back-scattered electron mode providing a clear view of the numerous holes on the wrinkled fiber wall. These pores facilitate mass and charge transfer across a larger number of specific surface areas, which are considered active

sites during electrochemical test (Supplementary Fig. 3b–e). Additionally, cross-sectional observations of individual mesh fibers or clusters reveal that the crumpled surface of the fibers creates extra pathways for simultaneous mass and charge transfer, including inner-fiber and inter-fiber pathways. This takes advantage of the ultra-porous

nature of the structure (Fig. 1e). At the interface between the mesh-structured electrode and the electrolyte, the mesh fibers form a shell-like structure that integrates with the dense electrolyte, with an appropriate amount of PNC73 powder particles serving as a homogeneous adhesive in between (Supplementary Fig. 3f). Additionally,

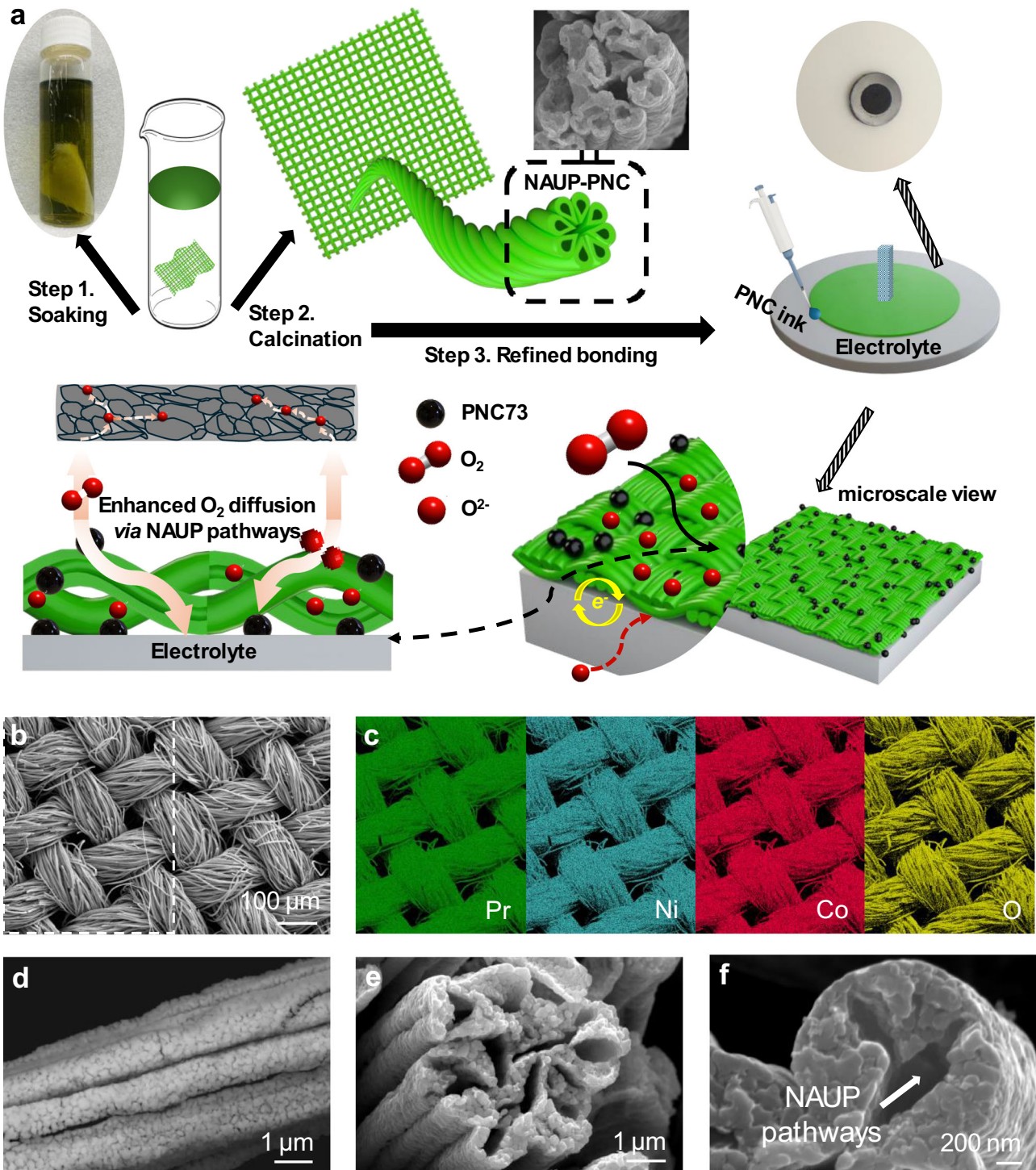

**Fig. 1 | Scalable fabrication and characterization of nano-structured ultra-porous PNC73 oxygen electrode. a** Schematic illustration of the process to fabricate NAUP-PNC73 mesh-structured electrode and its integration into the $BaCe_{0.4}Zr_{0.4}Y_{0.1}Yb_{0.1}O_{3-\delta}$ (BZCYYb4411[40])/BZCYYb4411-NiO half cells. The oxygen electrode has a diameter of 0.476 cm with a corresponding area of 0.178 cm², while the hydrogen electrode has a diameter of 0.872 cm with a corresponding area of 0.597 cm². **b** SEM image of mesh-structured NAUP-PNC73 (soaked in 1 M solution)

surface, displaying visible channels for charge and mass dual-transfer. **c** EDS mapping for constituent substances on selected mesh area, including Pr, Ni, Co and O. **d** SEM image (back-scattered electron mode) of a single mesh fiber, showing observable holes for dual-transfer (side view). **e, f** SEM images of a single mesh fiber, indicating apparent pathways for dual transfer (cross-sectional view) and semi-NAUP pathways in nanoscale.

X-ray diffraction (XRD) analysis of the two mesh-structured PNC73 electrodes and the powder-packed PNC73 electrode confirmed that all samples exhibit a pure crystalline phase (Supplementary Fig. 4).

Due to the self-assembly process, PNC73 particles within the fiber walls aggregate from the liquid phase to achieve a nanoscale size of less than 400 nm, which can be precisely controlled through loading and calcination temperatures. This particle size is optimal for enhancing activity in both the oxygen reduction reaction (ORR) in fuel cell mode and the oxygen evolution reaction (OER) in electrolysis mode (Fig. 1f). Considering its morphology and porosity, this innovative structure is termed the nano-architecture ultra-porous PNC73 (NAUP-PNC73) electrode. The expanded NAUP channels improve oxygen diffusion, providing high gas partial pressures at the interfaces and thereby facilitating reaction kinetics on this nano-structured electrode.

To investigate the porous nanostructure of the NAUP-PNC electrode, transmission electron microscopy (TEM) was utilized to analyze its microstructure and atomic-level crystal structure. The high-resolution TEM (HRTEM) image of PNC nanoparticles within the mesh fiber, captured near their respective zone axes, is shown in Fig. 2a. The magnified view of the marked region (inset) reveals atomic arrangements consistent with the [111] zone axis of orthorhombic per-ovskite symmetry. The measured interplanar spacing of 0.339 nm vali-dates the lattice parameter, aligning with the pure PNC phase index information. To further support these observations, a simulated atomic arrangement of the PNC phase is provided, highlighting the atomic pattern. The selected area electron diffraction (SAED) results in Fig. 2b corroborate the specific planes of the PNC crystalline structure. Addi-tionally, a high-angle annular dark-field (HAADF) TEM image, along with elemental distribution maps, is presented in Fig. 2c and d. These maps confirm the uniform distribution of all elements comprising the PNC phase within the selected region (highlighted by the orange rectangle in Fig. 2c). Through the line scan analysis (Figs. 2e–i) and point analysis (Supplementary Fig. 5) of a single NAUP fiber in HAADF mode, the ele-ments Pr, Ni, and Co exhibit similar compositional peaks with varying amplitudes along the entire fiber. This consistency indicates that the

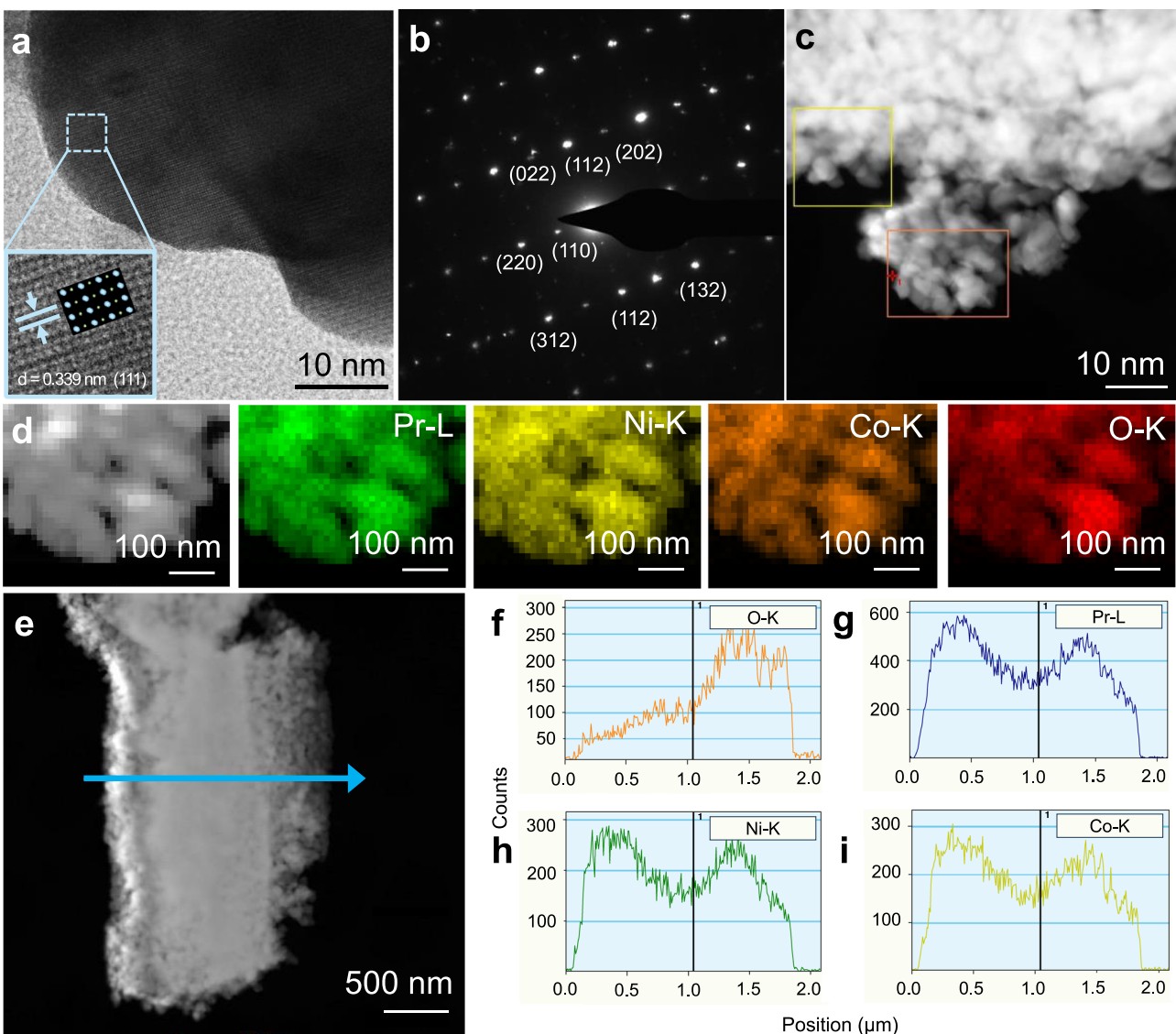

**Fig. 2 | TEM characterization of NAUP-PNC oxygen electrode. a** HRTEM image of NAUP-PNC nanoparticles. The inset contains magnified view of the marked region with a simulated atomic model of the PNC phase (orthorhombic perovskite structure, *Pnma* space group). **b** SAED pattern of NAUP-PNC nanoparticles. **c** HAADF image of NAUP-PNC nanoparticles; the representative areas are squared. **d** EDS elemental maps of the NAUP-PNC electrode confirm the presence of the PNC phase. From left to right, the selected area for EDS (orange rectangle in Fig. 2c) shows praseodymium as a green pattern, nickel as a yellow pattern, cobalt as an orange pattern, and oxygen as a red pattern, arranged in order. **e** EDS line scan (blue arrow) of a single NAUP-PNC electrode fiber. **f–i** Elemental information of line scan, oxygen-orange (**f**); praseodymium-blue (**g**); nickel-green (**h**) and cobalt-yellow (**i**).

prepared NAUP mesh retains uniform elemental composition and is well-suited for fabrication onto half cells.

## Synergistic effects of enlarged dual-transfer pathways and enhanced interfacial bonding on reduced polarization resistance of $R_o$ and $R_p$

Electrochemical performances of PCECs are closely related with the polarization resistances from each cell component and ohmic resistance ($R_o$) and polarization resistance ($R_p$) are two crucial parameters for evaluating the materials or microstructure. Thus, significant advancements have been achieved to reduce the resistance mainly by microstructure optimization[28–30].

To rationally study the synergistic effect of this NAUP electrodes on the polarization resistances, the impedance spectra from full cells are systematically investigated to analyze the relationship between the refined electrode microstructure and improved interfacial activity. As clearly seen in Figs. 3a–3c, the $R_o$ value dropped below 0.20 $\Omega$ cm$^2$, e.g., 0.190 $\Omega$ cm$^2$ under open-circuit voltage (OCV) conditions and 0.158 $\Omega$ cm$^2$ in electrolysis mode at 1.30 V and 600 °C, respectively. Compared with the conventional powder-packed PNC73 electrode, both $R_o$ and $R_p$ of this NAUP electrode are significantly reduced. For example, the $R_p$ of NAUP-PNC73 in full cell was only 40% of that observed in powder-packed PNC73 electrode. A consistent trend was witnessed across the entire temperature range with exact resistance values and corresponding standard deviations shown in Supplementary Fig. 6. Furthermore, it is evident that the values of $R_o$ and $R_p$ presented under steam electrolysis were much lower than those collected under OCV conditions from the identical cells. This behavior is well explained by the distinct overpotential conditions, the accumulation of reactant concentrations on both electrodes, the efficient utilization of generated heat, and the maintenance of effective dual-transfer pathways. In electrolysis mode, the higher applied voltage reduces polarization resistance, enhances ion conductivity, and improves charge transfer, aided by heat generated during steam electrolysis[31,32]. The heat effect aligns with the observed results (0.192 $\Omega$ cm$^2$ at OCV and 0.158 $\Omega$ cm$^2$ at 1.30 V for $R_o$, 0.073 $\Omega$ cm$^2$ at OCV and 0.053 $\Omega$ cm$^2$ at 1.30 V for $R_p$). This phenomenon is probably due to the enhanced proton conduction and localized Joule heating, further lowering resistance and improving the efficiency of NAUP-PNC electrodes.

To gain a deeper understanding of the elemental reaction steps in polarization resistance, a distribution of relaxation times (DRT) analysis was performed to deconvolute their relationship. By separating the three impedance arcs and using appropriate fitting parameters, we can reveal the underlying electrochemical processes (Fig. 3d). The peak intensities in the primary frequency regions—low-frequency (LF), intermediate-frequency (MF), and high-frequency (HF)—are quantified in terms of their statistical variability, corresponding to bulk ionic conduction, gas diffusion and interfacial reaction process[33]. It was observed (Fig. 3e) that the NAUP-PNC73 electrode under FC and EC modes demonstrates smaller intensities than the conventional electrode structure in the LF region, which clearly indicated that the mass transfer was significantly enhanced by the nanoscale ultra-porous structure. Similarly, the shrinking peaks (green and orange) in the MF region further confirmed the optimized oxygen surface exchange featuring faster and more efficient oxygen adsorption, dissociation and incorporation on the interface. This behavior was more pronounced in EC mode, where a 1.30 V electrolysis voltage was applied during measurement. In the HF region, the lower intensity observed in EC mode indicated enhanced charge transfer with fewer limitations due to the dual-transfer channels of the NAUP structure. This effect was not as evident in FC mode. It is important to note that the suppression of HF peaks aligns with the unobstructed pathways for charge movement during steam electrolysis, as illustrated in Fig. 3c.

The activation energy is the minimum amount of energy required for this electrochemical reaction to proceed, representing the energy barrier that reactants must overcome during ORR or water splitting reaction, which is related to the performance[34,35]. As depicted in Figs. 3f–3g, after combining the initial results of $R_o$ and $R_p$ noted in Supplementary Fig. 6a-b with Arrhenius Equation for activation energy (Supplementary Eq. 1), the diagram of ln $R_{o/p}$ versus 1000 $T^{-1}$ was plotted along with fitted straight line, clearly exhibiting the activation energy values. The calculated $E_{a,o}$ of conventional powder-packed PNC cells at fuel cell mode is 0.114 eV, whereas that of two representative NAUP-PNC cells, whether acquired at open circuit condition or 1.30 V, is around 0.106 eV. Considering the polarization resistance as major contributor to the overall impedance, 0.329 eV, 0.304 eV, 0.308 eV, and 0.308 eV (only first value is from the conventional powder-packed PNC cell). Considering polarization resistance as the major contributor to overall impedance, the $E_{a,p}$ is 0.329 eV for the conventional powder-packed electrode, 0.304 eV for NAUP-PNC electrode under open circuit, and 0.308 eV under 1.30 V. The reduction of activation energy is clearly attributed to the enhanced reaction kinetics, which is more compared in Fig. 3h, Supplementary Tables 3-4. $E_{a,p}$ is strongly influenced by the nanoscale properties of the catalyst. Nanometer-sized catalysts provide a significantly higher surface area-to-volume ratio, increasing the number of active sites available for reaction. This enhanced interaction between reactant molecules and the catalyst surface improves reaction efficiency and lowers the apparent activation energy. As revealed by TEM analysis (Fig. 2), the nanoparticles in the NAUP-PNC electrode, measuring tens of nanometers, exhibit quantum effects and altered surface electronic properties. These characteristics stabilize transition states and reaction intermediates, further reducing the energy barrier. On the other hand, the optimized porosity and pore connectivity in the NAUP electrode design enhance mass transfer pathways and minimize diffusion resistance. Within this optimized framework, a slight but consistent reduction in activation energy was observed, with a 7.37% decrease for $E_{a,o}$ and a 7.68% decrease for $E_{a,p}$. These reductions underscore the effectiveness of the NAUP electrode in stabilizing reaction intermediates and improving the adsorption and desorption efficiency of intermediates (Supplementary Fig. 7). The ultra-porosity and expanded active sites of the NAUP electrode facilitate rate-determining steps for both the oxygen reduction reaction (ORR) and oxygen evolution reaction (OER), resulting in enhanced catalytic activity and superior electrochemical performance.

To understand the synergistic mechanism behind the reduced overall resistance and activation energy, the cross-sectional view of a NAUP-PNC full cell was initially analyzed (Fig. 4a). The cell configuration with the hydrogen electrode, featuring well-distributed holes, a dense and flat electrolyte, and the intricately intertwined mesh-structured NAUP oxygen electrode, is clearly visible. Although delamination is a potential concern due to the differing structures and morphologies of the mesh electrode and electrolyte, they are effectively bonded together with the aid of the adhesive PNC ink. Here, the NAUP-PNC full cells achieved the unique "homogeneous composition but heterogeneous morphology" compatibility, ensuring the adequate physical and chemical stability. The utilized PNC ink acted as the agent in microscale to better couple the large-grain electrolyte of tens of micrometers with NAUP-PNC oxygen electrode of nanometers (Fig. 4b). Assuredly, the mesh-structured electrode facilitates the massive gas transport and efficient catalysis via the gas channels and nanoparticles on the fibers. Further throughout PNC ink, oxygen ions realize bulk diffusion and incorporation along interfaces via enlarged dual-transfer pathways. Figures 4c and d presents the tighter-bonding areas along the newly created microscale and nanoscale PNC interface, which are PNC particles ($\alpha$) and NAUP-PNC structure ($\beta$). The bridge-shaped NAUP-PNC plays a crucial role in both mass transport and charge transfer. The thickness of this hybrid electrode has been reduced to around 10 μm, which is the minimum value identified for maintaining sufficient structural strength[36,37]. While a thinner electrode

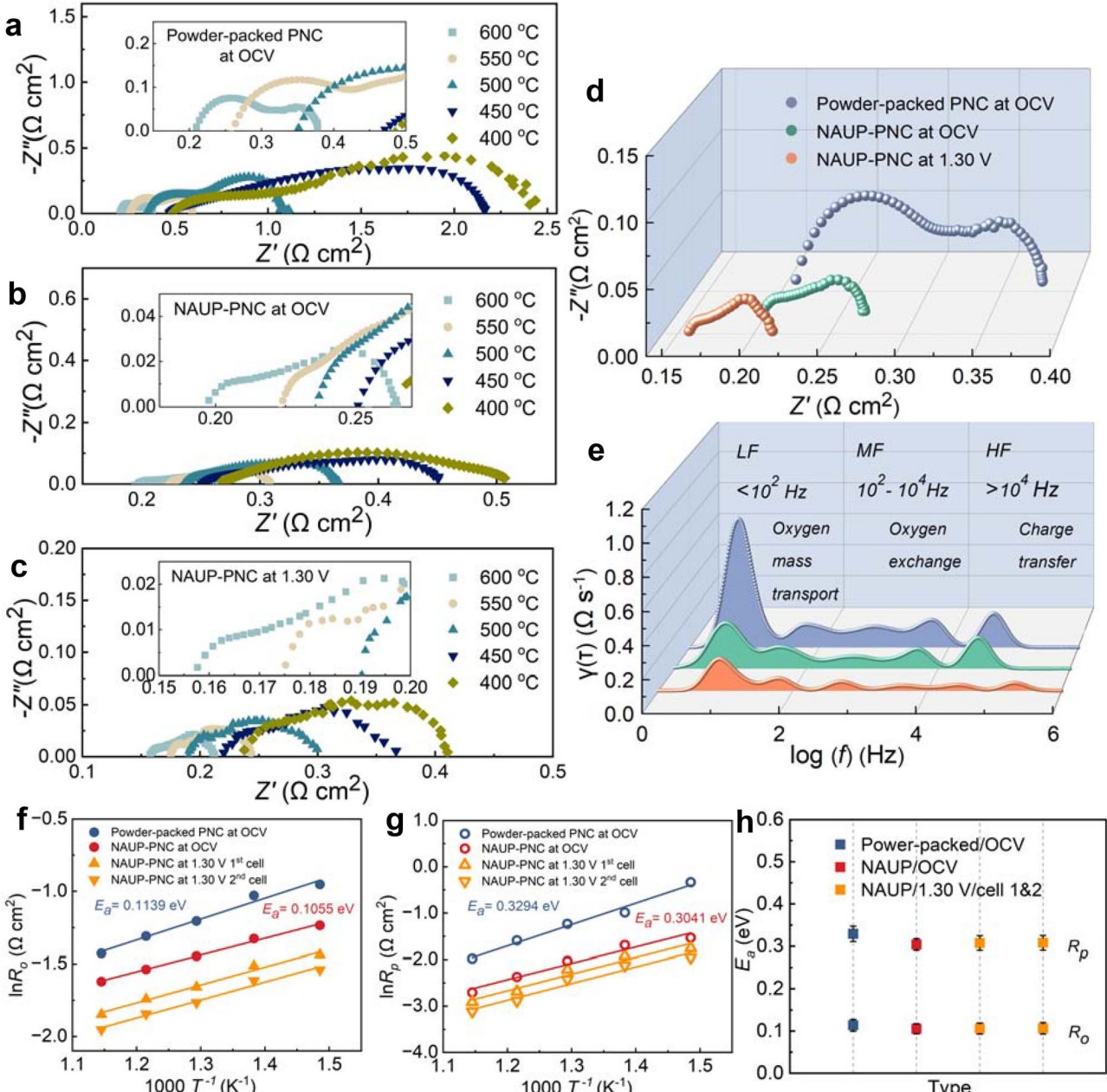

**Fig. 3 | Correlated elucidation on synergistic impacts of electrode structures via electrochemical impedance spectra and analysis of distribution of relaxation times (DRT) and activation energy. a** EIS spectra of powder-packed PNC in full cells measured under OCV conditions at 600 °C. Hydrogen (20 standard cubic centimeter per minute, sccm) and oxygen (40 sccm) were respectively fed into the cells. **b** EIS spectra of NAUP-PNC cells measured under OCV conditions at 600 °C with the same gas conditions. **c** EIS spectra of NAUP-PNC cells measured at 1.30 V and 600 °C under electrolysis conditions (H$_2$, 20 sccm, was fed to hydrogen electrode side and 20 vol.% H$_2$O + 80 vol.% O$_2$, 40 sccm was fed to steam electrode). **d** Comparison on EIS spectra from three representative conditions: regular powder-

packed PNC cells at OCV condition; NAUP-PNC cells at OCV, and NAUP-PNC cells at 1.30 V, displaying reduced ohmic and polarization resistances. **e** The corresponding DRT analysis of the EIS spectra shows that all peaks are smaller across the entire frequency range. **f, g** Arrhenius plot of ohmic resistance and electrode polarization resistance of cells under four characteristic conditions with fitted straight lines (two cells were used to validate the repeatability). **h** The corresponding activation energies for $R_o$ and $R_p$. Each test was conducted for six times to get the statistical standard deviations, presented as error bars. *All voltage values reported in this work was not iR corrected. Source data for Fig. 3 are provided as a Source Data file.

could potentially reduce the transfer efficiency, this has been largely offset by the increased specific surface area created by its ultra-porous structure. To explore the relationship between dual-transfer efficiency and pore size in NAUP electrodes, meshes were prepared at calcination temperatures of 650 °C, 750 °C, and 850 °C, denoted as NAUP-650, NAUP-750, and NAUP-850, respectively. This allowed visualization of the impact of calcination temperature on dual-transfer pathways. As shown in the SEM images of temperature-dependent meshes (Fig. 4e),

the lower calcination temperature (650 °C) preserved finer structures with smaller particle sizes due to reduced sintering and minimized grain growth, resulting in larger pore sizes that enhance gas diffusion. In contrast, higher calcination temperatures reduced pore size due to enhanced sintering and crystallinity, which led to larger particle sizes and decreased surface area, as observed in the NAUP-850 mesh. To balance pore size and active PNC nanoparticle content for optimal reaction facilitation, electrochemical tests were conducted using

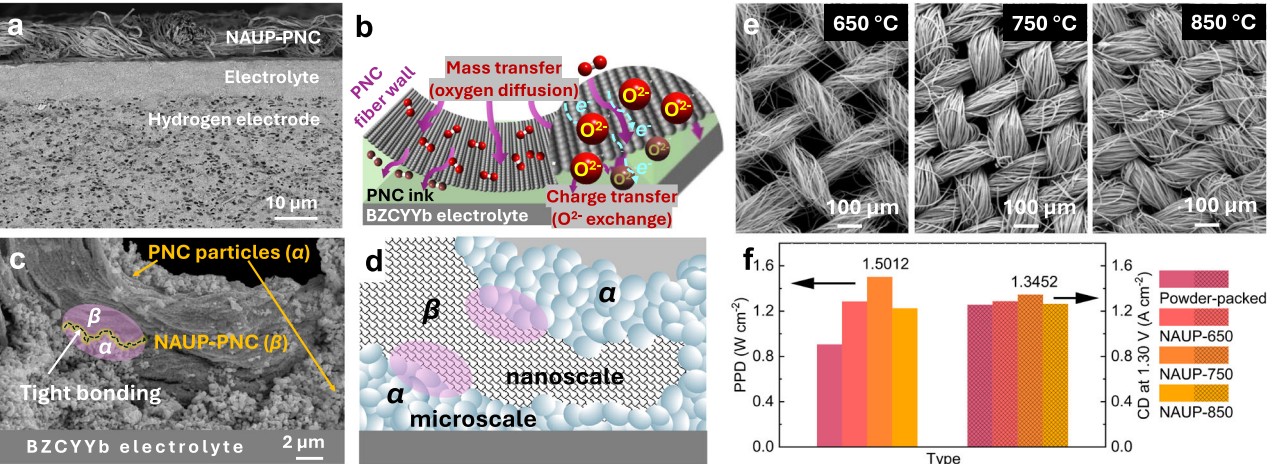

**Fig. 4 | Illustration of the integrated NAUP PNC electrode and hybrid structure for optimal dual mass/charge transfer and interfacial strength in PCEC. a** SEM image of cross section of as-fabricated NAUP-PNC/BZCYYb4411/BZCYYb4411-NiO "sandwiched" structure. Despite the use of a mesh-structured electrode, a strong and secure bonding between the electrode layer and electrolyte is still seen, with no evident signs of delamination. **b** Schematic illustration of occurrence of charge and mass dual-transfer through the single NAUP-PNC fiber inner wall to slight loading of PNC glue ink, ultimately to the electrode/electrolyte interface under the fuel cell mode. During the former fabrication, 10 µl PNC73 ink for certain electrode area (0.178 cm²) was used to achieve refined bonding (Fig. 1a, Step 3). **c** SEM image of single mesh fiber section, verifying the tight bonding between PNC73 particles ($\alpha$) and NAUP-PNC structure ($\beta$). **d** Schematic illustration of enhanced bonding existing between $\alpha$ (microscale) and $\beta$ (nanoscale). **e** SEM images of mesh calcinated at 650 °C, 750 °C and 850 °C. **f** Bar chart of the comparison on the PPD and CD at 1.30 V at 600 °C for different meshes. Source data for Fig. 4 are provided as a Source Data file.

single full cells fabricated with meshes of the three porosities (Fig. 4f). Compared to the conventional powder-packed PNC electrode, all NAUP-PNC electrodes demonstrated improved performance. Notably, the NAUP-750 achieved a peak power density improvement of up to 66% at 600 °C and higher current density for electrolysis at 1.30 V. These results highlight the enhanced performance of the NAUP-750 mesh structure, attributed to its optimal porosity and dual-transfer efficiency. This NAUP-PNC mesh-structured electrode is designed to improve interfacial properties, resulting in lower activation energy for ORR/OER and reduced overall resistance, particularly $R_p$. This synergistic enhancement is a key highlight of its performance.

## Superior electrochemical performances in fuel cell and steam electrolysis modes

To investigate the NAUP-PNC full cell performance, the *I-V-P* curves in fuel cell mode and polarization curves in electrolysis mode have been systematically measured, respectively. We fabricated two cells: one with a conventional powder-packed electrode and the other with a NAUP electrode, to highlight the improvements achieved with the new structure. For both cells, the open-circuit voltages (OCVs) of the cells were above 1.03 V, indicating a dense electrolyte membrane and confirming the absence of gas leakage (Figs. 5a–5b). Conclusively, the as-fabricated NAUP-PNC full cells attained an average increment of 58% on peak powder density (PPD) than that of powder-packed PNC full cells over the entire set temperature range (Supplementary Table 5). At 600 °C, the PPD was optimized from 0.906 W cm⁻² to 1.501 W cm⁻² with H₂ (20 sccm) and O₂ (40 sccm) fed to the electrodes, which incredibly towered over the results in latest reported literatures (Supplementary Table 1). With the test temperature set as 550 °C, 500 °C, 450 °C and 400 °C, the PPD yield shows improved performance by 59%, 60%, 67% and 37%, respectively (Supplementary Fig. 8). It strongly presents the excellence of NAUP mesh structure operated at low temperatures. When operating NAUP-PNC cells under steam electrolysis for hydrogen production, favorable performance was similarly demonstrated with the contrast to the current densities of powder-packed PNC electrode at 400 °C–600 °C. At 600 °C, the electrolysis current density reached 5.04 A cm⁻² at 1.60 V, however, the conventional PNC cells had only 3.36 A cm⁻²,

highlighting the 150% enhancement for electrolysis (Figs. 5c–5d, Supplementary Table 6). Additionally, current densities at other four tested temperatures till 400 °C all increased by over 32%, demonstrating significantly better performance in NAUP-PNC full cells compared to powder-packed PNC full cells under the same applied voltages. The aforementioned advancements have validated the enhanced electrode kinetics and reduced total resistance from the NAUP nanostructure, clearly indicating the significant boost of the investigated performances in both fuel cell and electrolysis modes among the recent PCEC progresses.

To evaluate the dependences of PCEC conditions on electrochemical performances, the steam and oxygen partial pressures are varied to find how these parameters may affect hydration and other defect formation within the electrolyte to consequently influence electrolysis behaviors. Firstly, a slight increase in electrolysis current density is observed with rising steam pressure (Fig. 5e). Specifically, the current density improved from 4.03 A cm⁻² at 1.60 V with 3% H₂O to 5.04 A cm⁻² with 20% H₂O. This behavior contrasts with that of conventional electrode structures, which do not exhibit a positive effect from increased steam pressure. This improvement is attributed to the enhanced gas diffusion and surface kinetics of the NAUP electrode. Additionally, higher steam pressure facilitates greater hydration in the triple-conducting PNC electrode, thereby enhancing both surface reaction efficiency and bulk proton conduction throughout the electrode layer and interface. However, further increase to 30% H₂O did not improve the current density anymore but slightly dropped to 4.74 A cm⁻² (Supplementary Table 7). Secondly, by increasing the oxygen sweep gas from 10 to 60 sccm, the electrolysis current density also showed the uptrend and reached equilibrium (Fig. 5f), which further validated that the NAUP structure allows large steam flow rate that enhances electrolysis activity. When comparing the performance of this work with other recent and representative studies, the current density and PPD of NAUP-PNC cells exhibit significant improvements, whether in fuel cell mode or steam electrolysis (Supplementary Fig. 9). All reference data were gathered from the tests using the same experimental gas conditions (H₂/O₂) as our study to reinforce the comparison (Supplementary Table 8). For example, the PPD from the cell with microwave-assisted sintered

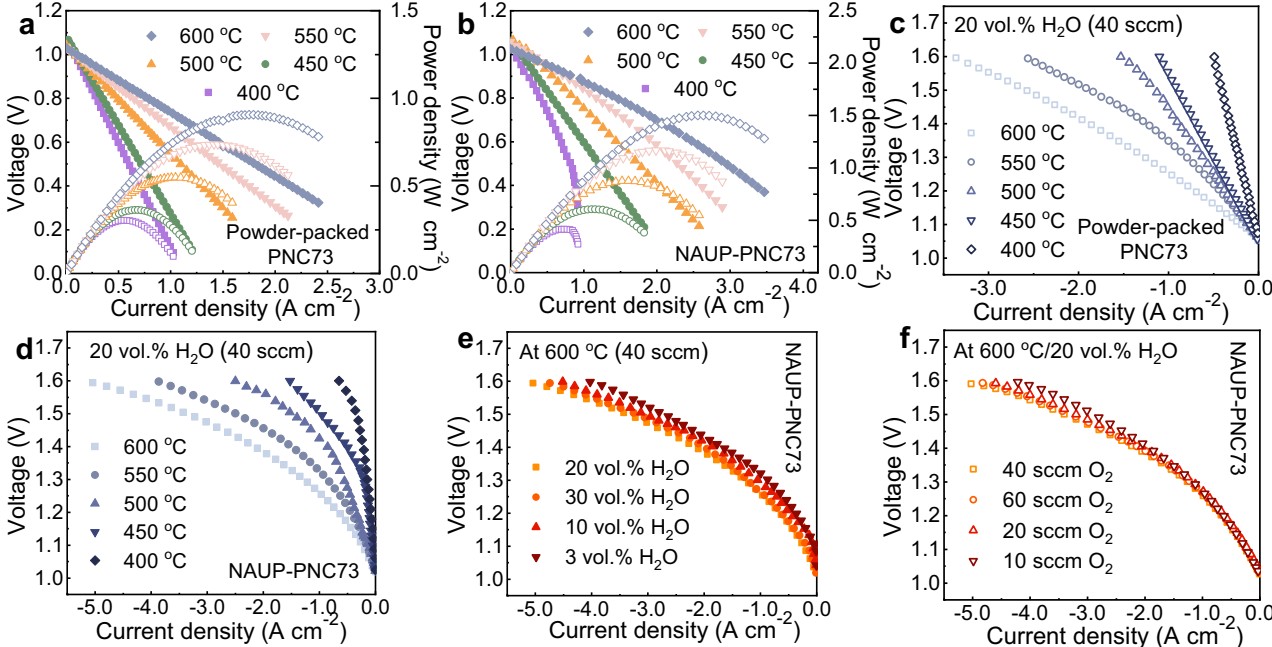

**Fig. 5 | Electrochemical performances of NAUP-PNC73 full cells in both fuel-cell and electrolysis modes as compared to full cells with conventional powder-packed electrodes. a** I-V-P curves of the cell with a conventional powder-packed PNC73 electrode measured at 400–600 °C. **b** I-V-P curves of a representative NAUP-PNC73 cell at 400–600 °C. **c** Polarization curves of a powder-packed PNC73 cell in electrolysis mode at 400–600 °C. $H_2$ (20 sccm) and 20 vol.% $H_2O$ + 80 vol.% $O_2$ (40 sccm) was fed to the electrode. The applied voltage for all tests ranged from 1.60 V to OCV. **d** Polarization curves of a NAUP-PNC73 cell in electrolysis mode at 400–600 °C. **e** Comparison of electrolysis performance of NAUP-PNC73 in terms of different vol.% $H_2O$ in $O_2$ at the flow rate of 40 sccm and 600 °C. **f** Comparison of electrolysis performance of NAUP-PNC73 in terms of different flow rate of $O_2$ at 600 °C. All tests conducted were under the same relative humidity of $O_2$, which was 20 vol.% $H_2O$ in $O_2$. Source data for Fig. 5 are provided as a Source Data file.

BCFZY electrode achieved 0.88 W cm⁻² at 600 °C. While this demonstrates improved performance under extreme sintering conditions, it still does not meet the benchmark of 1.0 W cm⁻² [14]. Notably, the NAUP-PNC electrode in this work surpassed 1.50 W cm⁻², same at 600 °C, marking at least a 32% increment compared to other referenced studies. In terms of steam electrolysis performance, the current density at 1.30 V for NAUP-PNC full cells significantly exceeds that of other studies, showing approximately a 1.8-fold improvement. Specifically, the $(PrBa_{0.8}Ca_{0.2})_{0.95}Co_2O_{6−δ}$ (PBCC95) oxygen electrode exhibited 0.71 A cm⁻² at 1.30 V and 600 °C owing to the A-site cation ordered and deficient material, whereas the current density achieved in this work was -1.35 A cm⁻² under identical conditions, ensuring it is highly competitive for large-scale reversible operations. Overall, cell performance may be slightly affected by relative humidity or oxygen flow rate, with current densities decreasing by up to 20%. This marginal impact further confirms the robust physical and chemical stability of the nanostructured NAUP-PNC electrode. Faradaic efficiency (FE) and energy efficiency (EE) were evaluated for NAUP-PNC full cells to demonstrate their high performance in energy-intensive processes, such as water splitting. During steam electrolysis under 20 vol.% $H_2O$ at 600 °C, the cells achieved an impressive FE exceeding 95% at a current density of 0.6 A cm⁻². Additionally, the energy efficiency for converting electrical energy into chemical energy surpassed 80% under the same conditions (Supplementary Fig. 10). These results highlight the feasibility of hydrogen production using the NAUP-PNC oxygen electrode.

## Electrochemical performance durability in various degradation measurements

The electrode composition used in this study has been validated as a promising TCO material, free of alkaline earth metals, and exhibits significantly enhanced proton conductivity due to its readiness for hydrating lattice structure to form reasonable proton defects. The

electrochemical durability of this electrode in fuel cell mode has recently been demonstrated through an accelerated transient test [17,18]. When the conventional electrode is transformed to incorporate the NAUP structure, its performance durability under various transient conditions is systematically examined. Initially, the full cell was operated at a constant voltage of 1.30 V in 20% $H_2O$ at 600 °C for 50 hours, followed by a switch to 1.40 V for an additional 55 hours. The resulting current density exhibited only a 1.03% decrease during the first electrolysis period and a 1.24% decrease during the second (Fig. 6a, Supplementary Table 10). During the test, EIS measurements at 1.30 V were recorded every 10 hours to monitor changes in polarization resistance (Fig. 6b). Notably, the value of $R_o$ remained stable at approximately 0.15 Ω cm², while $R_p$ was maintained at around 0.05 Ω cm², with an acceptable fluctuation of about 5% (Supplementary Table 9). To assess the feasibility of operating in reversible modes, the cell was tested by alternating the voltage between 1.3 V and 0.7 V, and the response in current density was observed (Fig. 6c). Throughout the 5 cycles, the current density responded immediately to voltage changes, demonstrating that the PNC electrode and its structure can swiftly adapt to mode transitions. The transient degradation rates were 1.87% for steam electrolysis and 1.55% for fuel cell operation (Supplementary Table 11), indicating that NAUP-PNC cells are both reliable and flexible as reversible protonic solid oxide cells.

Escalated transient testing is a valuable approach for assessing the resilience of cell properties by examining the adaptability of materials and interfaces under dynamic conditions. NAUP-PNC full cells were subjected to step-voltage and time-interval-based transient tests during steam electrolysis to gain deeper insights into the degradation mechanisms. In the step-voltage testing, the cell was cycled among 1.50 V, 1.35 V, and 1.20 V, with each voltage applied for 20 minutes (Fig. 6d). The electrolysis current density swiftly and consistently transitioned between 3.30 A cm⁻², 1.70 A cm⁻², and 0.75 A cm⁻², aligning with the initial performance measurements of the NAUP-PNC

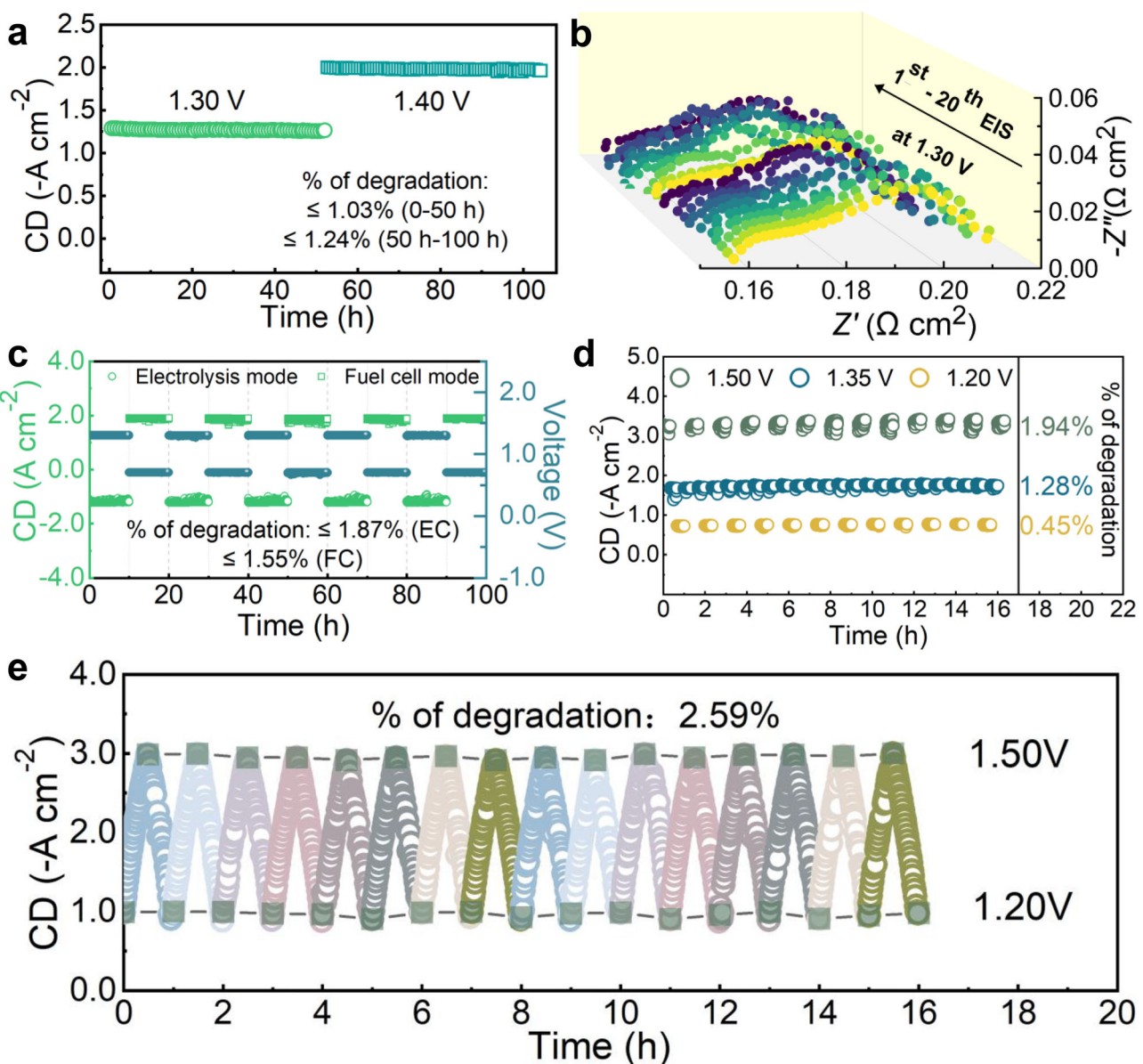

**Fig. 6 | Extraordinary stability and durability on accelerated stress tests of NAUP-PNC full cells under both fuel cell and electrolysis modes. a** Long-term stability tests for 100 hours, during which a single switch of applied voltage was made from 1.30 V to 1.40 V, constant electrolysis testing at a fixed voltage of 1.30 V for 50 hours, followed by 1.40 V for approximately 55 hours, showing degradation rate of current density under electrolysis mode was less than 1.25%. **b** Corresponding EIS spectra collected at 1.30 V for 20 times during the long-term stability test. **c** Reversible tests between fuel cell mode at 0.70 V and electrolysis mode at 1.30 V for 100 hours (5 cycles). **d** Transient tests (step-voltage based) under

electrolysis mode at 1.50 V, 1.35 V and 1.20 V respectively, dynamically operated across different modes. **e** Transient tests (time-interval based) under electrolysis mode between 1.20 V and 1.50 V. A cycle (from 1.20 V to 1.50 V, and then back to 1.20 V) lasted for 1 hour, dynamically operated across different modes. All tests were conducted at 600 °C. Fuel cell mode was operated using $H_2$ (20 sccm) for the anode and $O_2$ (40 sccm) for the electrode. Electrolysis mode was operated using $H_2$ (20 sccm) for the anode and 20 vol.% $H_2O$ + 80 vol.% $O_2$ (40 sccm) for the electrode. Source data for Fig. 6 are provided as a Source Data file.

electrodes. Importantly, degradation at 1.20 V was minimal, at only 0.45%, while other degradation rates were below 2% (Supplementary Tables 12-14). Despite the high overpotential during steam electrolysis causing minor damage to the nanostructure of the electrode layer, the NAUP-PNC full cells proved robust, maintaining reliable current density responses during rapid voltage changes.

In addition to step changes, the applied potential was varied based on specific intervals between 1.20 V and 1.50 V. The voltage was first increased and then decreased back to the original level, with each voltage-changing cycle lasting 1 hour and repeated 16 times. Figure 6e illustrates a clear periodic pattern in the current response with minimal deviations, and the statistical degradation rate was

2.59% (Supplementary Tables 15-17). This significant durability of NAUP-PNC full cells highlights their potential for industrial-scale applications.

### Structural resilience of the NAUP electrode on thermal cycles

Typically, solid oxide cells undergo heating and cooling processes across their operational temperature range, particularly during start-up and shut-down cycles[38,39]. The PNC material system has been elucidated to excel in the thermal stability during the multi-transient-cycling examination due to its improved thermal expansion behavior via the stoichiometric balance on various properties. In view of the frequent temperature fluctuations in realistic operation, it is crucial to

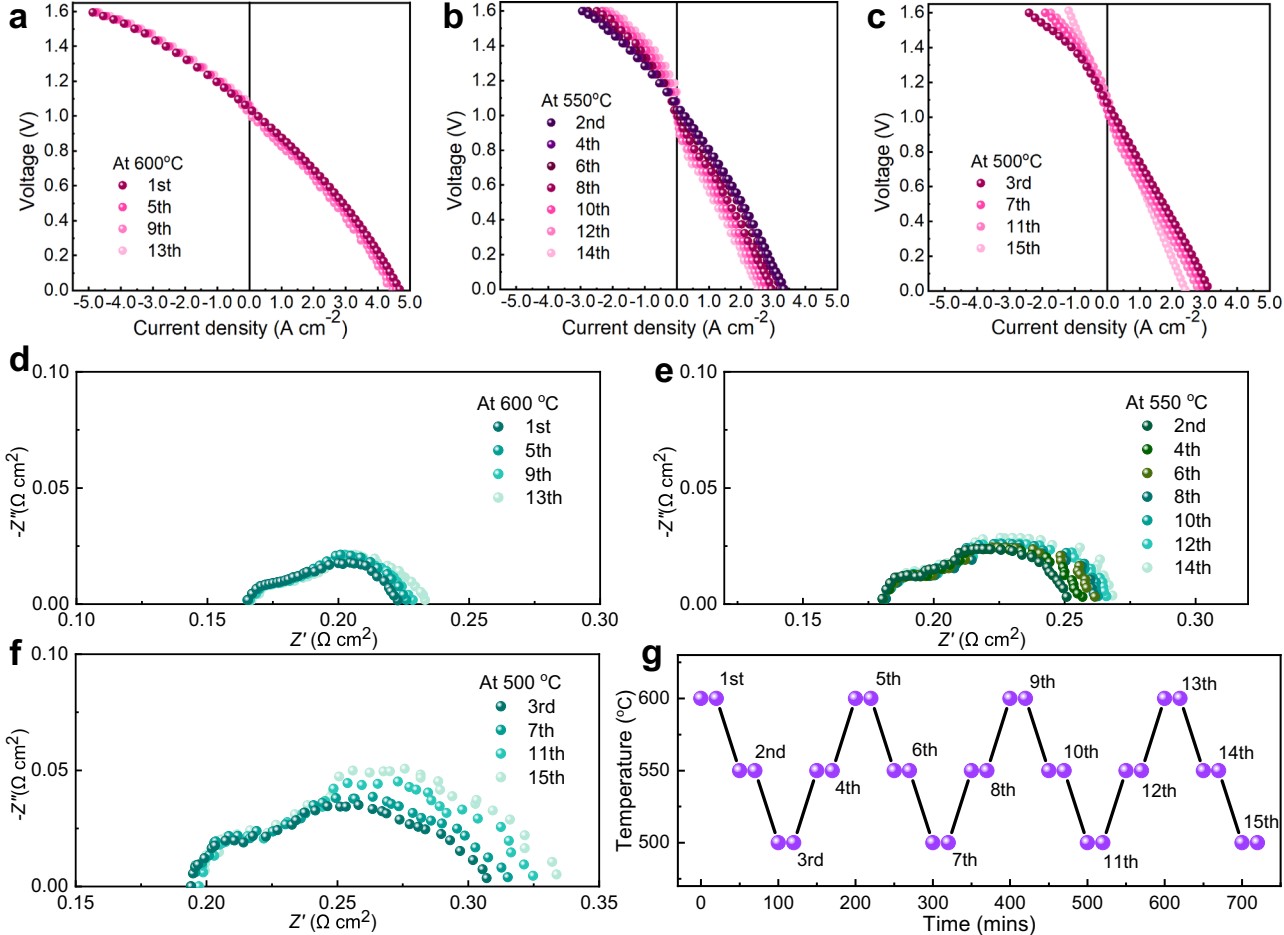

**Fig. 7 | Thermal cycle stability evaluation of NAUP-PNC electrode under both fuel cell and electrolysis modes, explicated by dynamically operated across different temperatures. a–c** *I-V* curves of fast-changing thermal cycling tests at 600, 550, 500 °C, respectively. **d–f** Corresponding EIS curves at 1.30 V of fast-changing thermal cycling tests at 600, 550, 500 °C respectively. $H_2$ (20 sccm) was used for the anode and $O_2$ (40 sccm) for the electrode in fuel cell mode. Electrolysis mode was implemented for the anode using $H_2$ (20 sccm) and the electrode using 20 vol.% $H_2O$ + 80 vol.% $O_2$ (40 sccm). **g** Illustration of the complete thermal cycling process. The temperature ramp rate was 50 °C per 30 minutes. The measurement was completed after 20 minutes at the temperature point. Source data for Fig. 7 are provided as a Source Data file.

perform comprehensive assessment of NAUP structure durability under various transient thermal cycle conditions.

As shown in Figs. 7a–C, the transient polarization curves at three temperatures reveal distinct behaviors in both fuel cell and electrolysis modes, with higher temperatures enabling faster recovery of reaction activity. At 600 °C, current density degradation after four cycles is minimal−1.12% in electrolysis mode and 1.76% in fuel cell mode (Supplementary Table 18)−highlighting the durability of NAUP-PNC full cells under optimal conditions. This improved recovery at higher temperatures is possibly due to thermally activated processes, such as enhanced ionic and protonic transport. Proton conduction is more efficient at elevated temperatures due to reduced energy barriers, while the diffusion of oxygen vacancies and chemical re-equilibration further contribute to performance restoration. These effects align with the observed low degradation rates, confirming the benefits of thermal activation on durability and performance recovery at 600 °C.

The impedance spectra were collected in electrolysis mode to reveal the possible mechanism for the discrete behaviors. As can be seen in Fig. 7d measured at 600 °C, there is no observable change on ohmic resistance while the electrode polarization resistance increases from 0.057 $\Omega$ cm² to 0.065 $\Omega$ cm2 at 600 °C after several cycles, reflecting a modest rise of less than 14%. Similarly, there is no increase in ohmic resistance at 550 °C and 500 °C during thermal cycling;

however, the polarization resistance increased by 23.2% and 20.9%, respectively (Fig. 7e and Fig. 7f). More detailed data were recorded in Supplementary Table 19. Admittedly, the polarization resistance, which is the primary contributor to overall resistance, increased due to unstable temperature regions in the furnace cavity caused by rapidly fluctuating thermal conditions (Fig. 7g). This led to significant mechanical stress on the full cell configuration and weakened the adhesive PNC junction sites beneath the ultra-porous mesh structure. Similar negative effects are observed when using only electrode slurry to fabricate powder-packed electrode full cells. These results indicate that more studies can be carried out to further improve the interfacial mechanical strength based on this NAUP electrode.

## Discussion

Engineering of nanostructured electrode into PCECs is crucial to enhance effectiveness and reliability for electrochemical performances. The hurdles remain on how to compromise the complexity of synthesizing nanostructure, effective activity and gas transport, and durability on morphology and interfaces. The created NAUP structure in this study explicitly addresses the concerns on the synthesis scalability, porosity loss, structure instability, and interfacial bonding. This template-assisted one-step formation of the triple conducting electrode is straightforward to improve activity and durability simultaneously.

The structure perfection in nanoscale highly facilitates the mass/charge transfer and interfacial bonding along boundaries, resulting in reduced ohmic and polarization resistance and lowering energy barriers. The ohmic resistance dropped from 0.24 to 0.15 $\Omega$ cm$^2$, whereas the polarization resistance was reduced from 0.138 to 0.05 $\Omega$ cm$^2$. Moreover, after stressed tests for stability, the total resistance consistently remained at lower level within minor changes. These makes it achievable to obtain enhanced PPD from 0.9 to 1.5 W cm$^{-2}$ at 600 °C, witnessing the increment of 166% under fuel cell mode and improved current density at 1.60 V from 3.361 to 5.042 A cm$^{-2}$, of 150% boost under steam electrolysis, indicating the elevated hydrogen and power. Still notably, the NAUP-PNC cells exhibit the tolerance to the various humidities and flexible gas conditions.

When evaluating the potential of switching to the NAUP structure to enhance PCEC performance, durability is a critical factor. The improved interfacial bonding between the NAUP-PNC electrode and the electrolyte, facilitated by the PNC73 ink as a transitional medium, ensures the robustness of NAUP-PNC full cells during both transient cycling and long-term testing. It also shows good thermal stability, ensuring the potential to adapt the real cell operation conditions. The expanded dual-transfer pathways not only accelerate triple-conducting rates but also enable effective thermal management across both operating modes and reversible operations, preventing thermal fatigue. This durability and thermal stability make NAUP-PNC a promising choice for resilient protonic ceramic devices used in hydrogen production.

In summary, we present a straightforward method for producing the mesh-structured PNC73 fabric, followed by a conventional co-sintering process with the electrolyte-anode bilayer, to create a spontaneously formed nano-architecture ultra-porous electrode suitable for both hydrogen and power production. The NAUP-PNC73 full cells demonstrate enhanced performance in reversible operations, with higher peak power density, reduced overall resistance, and enduring durability, making them suitable for integration into advanced solid oxide systems. With the underlying mechanisms now understood, scaling down the electrode microstructure to the nanoscale with extensive porosity is expected to benefit other electrochemical materials and devices, such as alkaline fuel cells, electrolyzers, and biosensors.

## Methods

### Fabrication of NAUP-PNC73 mesh-structure electrode

The prepared cotton fiber (FabricLA) was cut into 5 × 5 cm$^2$ squares and soaked in the PNC73 precursor solution. To get the specified concentrations, 1 M and 2 M, dissolve the stoichiometric amount of PNC73 powder in the desired solvent (a 1:1 mixture of 95% ethanol and deionized water) and agitate for half an hour until a clear phase is obtained. The desired solution was kept at the transparent glass containers for later use. Regular stirring before each use. The as-prepared PNC73 powder was fabricated using following procedure. Specifically, stoichiometric amounts of Pr(NO$_3$)$_3$·6H$_2$O (99.9%, Alfa Aesar), Ni(NO$_3$)$_2$·6H$_2$O (99.9985%, Alfa Aesar), and Co(NO$_3$)$_2$·6H$_2$O (98 + %, Alfa Aesar) were dissolved in deionized water along with EDTA and citric acid to prepare a transparent precursor solution. The chelation process was carried out with a molar ratio of EDTA:citric acid:cations = 1:1.5:1, and a cation concentration of 0.02 mol L$^{-1}$ for this batch. The precursor solution was then magnetically stirred and heated until a viscous gel formed. The gel underwent further heating at approximately 200 °C, leading to self-ignition and the formation of powdery ash. This ash was transferred to a muffle furnace for calcination at 1000 °C for 5 hours, resulting in a crystalline perovskite phase of PNC73.

After fully immersing the fiber for 2 hours, it was calcined at 650 °C/750 °C/850 °C for 3 hours with the heating rate of 1 °C/minute and the cooling rate of 3 °C/minute in ambient air to get mesh-structure electrodes with varying porosity. Once the black textile mesh was removed from the muffle furnace, it was trimmed to a specific area of 0.178 cm$^2$ to serve as the actual electrode layer for later use.

### Integration of NAUP electrode to PCEC full cells

The full cells were fabricated using a mix of tape-casting and electrode-brushing or mesh-constructed techniques. The complete cells were prepared by first making the BCZYYb4411/NiO hydrogen electrode and electrolyte green tapes using the tape casting procedure. By sandwiching layers of electrode support and electrolyte membrane at 70 °C for 5 hours, half-cell green tapes were manufactured. To remove the organic solvents and binder, the cells were pre-sintered at 920 °C after being taken from the tapes. Finally, the electrolyte was solidified by sintering the cells for 5 hours at 1500 °C to get the half cells.

For NAUP-PNC73 full cells, 10 µl of PNC73 ink was used to affix the mesh-structured electrode onto the electrolyte/hydrogen electrode bilayer. The electrode ink was prepared by blending 5 g of as-prepared PrNi$_{0.7}$Co$_{0.3}$ (PNC73) powder, 0.5 g of binder (5 wt% V-006, Heraeus, dissolved in α-terpinol), and 1 g of dispersant (20 wt% Solsperse 28000, Lubrizol, mixed in α-terpinol) with about 15 mL of ethanol using a high-speed centrifugal mixer (Thinky). After about 20-minute operation, the slurry attained a semi-flow condition with an appropriate viscosity for later use. Following that, dry the cells in the over first and then they were subjected to the identical sintering process to acquire the NAUP-PNC73 full cells. For powder-packed PNC73 full cells, PNC73 ink was brushed onto the dense electrolyte layer. This resulted in fully functional cells with an active area of 0.178 cm$^2$. The cells were then subjected to a firing process at a temperature of 1000 °C for 2 hours.

### Cells assembly and electrochemical measurements

Both kinds of full cells were enclosed on the top of the lab-made testing fixtures (Supplementary Fig. 11) using Ceramabond™ 552 as sealant with the oxygen electrode facing upwards. A current collector made of silver mesh was utilized, with leads connected by silver wires. Once the cell was assembled, it was heated to 600 °C at a rate of 2 °C/minute. When the temperature reached 600 °C, with the continuous H$_2$ flow to the hydrogen electrode at the rate of 20 sccm, NiO was converted to metallic Ni. Upon reaching the equilibrium and stable OCVs were captured to indicate the complete reduction, 40 sccm O$_2$ was fed to the electrode to collect the data using the potentiostat (Parstat MC, Princeton Applied Research). Under fuel cell mode, the current-voltage-power curves, impedance spectroscopy at OCV for both NAUP-PNC cells and powder-packed PNC cells with variable frequency from 10$^6$ Hz to 0.01 Hz from 600 °C to 400 °C were noted. The amplitude was set at 50 mV RMS and the voltage range was +/−6V. For steam electrolysis, O$_2$ with certain relative humidity was introduced into the testing fixture through the custom-built humidifier. Different flow rates of oxygen were controlled by multiple flow meters. The polarization curves and impedance spectroscopy at 1.30 V from 600 °C to 400 °C were also recorded. Different relative humidities and flow rates of oxygen were considered to revalidate the electrolysis performance of NAUP-PNC cells. All electrochemical data were collected and process using VersaStudio software.

Comprehensive impendence analyses were conducted once the resistance data were collected. DRT analyses were completed through the Gamry Echem Analyst software while the activation energy was calculated using Arrhenius equation. Cells also went through long-term durability tests, transient cycling tests, reversible cycling tests and thermocycling tests. The long-term durability test for steam electrolysis was operated at 1.30 V for the first 50 hours and then at 1.40 V for the second 50 hours. Through the process, impedance curves were recorded for 20 times to verify the decreased total resistance. The reversible test was carried on at 0.70 V for fuel cell mode

and at 1.30 V for electrolysis for the total 100 hours. The transient cycling tests were maintained at 1.50 V, 1.35 V and 1.20 V for the step voltages while also varying between 1.20 V and 1.50 V for certain time intervals to witness current responses with any degradations. All mentioned experiments underwent at 600 °C. The thermal cycling tests were under 600 °C, 550 °C and 500 °C for a total 15-time assessment with the corresponding impedance curves presented. The acquisition of all data was performed using VersaStudio Software.

## Faradaic efficiency and energy efficiency measurements

To evaluate the electrolysis performance for hydrogen production, humidified oxygen gas was supplied to the NAUP electrode with a steam concentration of 20%. Pure hydrogen gas (40 sccm) was fed to the hydrogen electrode as the sweep gas. To evaluate the FE (%), the hydrogen electrode outlet gas was injected into a specially designed flow meter to quantify the hydrogen production rate. The FE (%) was calculated using

$$\nu = \frac{V_m}{2F} \times i$$

$$FE(\%) = \frac{\nu_e}{\nu} \times 100\%$$

where the theoretical hydrogen production flow rate, $\nu$ (sccm), is determined by several key parameters: the electric current, $i$ (A), which drives the reaction; the factor $2$, representing the two electrons transferred per molecule in the steam electrolysis process; the Faraday constant, $F$, valued at $9.6485 \times 10^4$ C·mol$^{-1}$; and the molar gas volume, $V_m$, set at 22,400 mL·mol$^{-1}$ under standard conditions. Meanwhile, the experimentally observed hydrogen production rate is denoted as $\nu_e$, providing a practical counterpart to its theoretical prediction.

The electrolysis energy efficiency was calculated according to

$$EE(\%) = \frac{FE(\%)}{\upsilon}$$

$\upsilon$ is the corresponding voltage (V).

## Characterizations

Scanning electron microscope (TFS Quattro S) was utilized to examine the surface of NAUP-PNC73 electrode and the cross-sectional area in both secondary electron and backscattered electron modes. Using the backscattered mode, detailed ultra-porous structure of mesh fiber was revealed. After acquiring the SEM images of certain regions, direct the images into Pathfinder X-ray microanalysis software to complete the EDS mapping to confirm all desired elements were successfully incorporated onto the mesh structure. The automatic peak identification, background subtraction and matrix corrections for quantitative analysis were acquired as a Spectral Imaging dataset. The distributed PNC elemental information exhibited contrast indicatives of the existence of praseodymium, nickel and cobalt, confirming the ideal compositional substances for later use. The morphology of the NAUP-PNC mesh was analyzed using transmission electron microscopy (TEM). The nanoparticles exhibited a uniform size distribution, ranging from approximately 20 to 100 nm. Energy dispersive X-ray (EDX) analysis, conducted with the TEM, confirmed the presence of Pr, Ni, and Co within the nanoparticles. A high-resolution TEM (HRTEM) image of the grain edge revealed a highly crystalline structure, corresponding to the (111) crystal plane of the perovskite structure, with a lattice inter-planar spacing of d = 0.339 nm. Furthermore, the selected area electron diffraction (SAED) pattern of the boxed region confirmed the long-range ordered crystal structure. A line-element scan demonstrated the homogeneous distribution of the three cations within the selected range. The crystalline phase structure of PNC73 powder and mesh structure were examined by X-ray diffraction (XRD, Rigaku SmartLab) using Cu rotating anode source.

## Data availability

All data needed to evaluate the conclusions in the paper are present in the paper and/or the Supplementary Information. Source data are provided with this paper. All other data supporting the findings of the study are available from the corresponding authors if asked for. Source data are provided with this paper.

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

## Acknowledgements
This work is supported by the US Department of Energy under contracts USDOE DE-EE0011336 and USDOE DE-FE0032235. H.D. acknowledges the startup research grant from the University of Oklahoma, and the authors gratefully acknowledge the financial support from the University of Oklahoma Libraries' Open Access Fund for publication.

## Author contributions
W.W., C.D. and H.D. conceptualized the work and methodology. Y.Z. and Z.Z. prepared the samples. S.Z. performed characterizations and measurements with data analysis and conducted all electrochemical tests. S.K. accomplished the data gathering. H.D. supervised the project and acquired funding. All authors contributed to the writing of the original draft and the review.

## Competing interests
The authors declare no competing interests.
