## [Transparent Peer Review file · Nature Communications]

Enhancing Surface Activity and Durability in Triple Conducting Electrode for Protonic Ceramic Electrochemical Cells

Corresponding Author: Professor Hanping Ding

Version 0:

Reviewer comments:

Reviewer #1

(Remarks to the Author)

Zheng et al. present a study on an enhanced cathode material for protonic ceramic electrochemical cells (PCECs), developed using a nano-fiber-structured ultra-porous design approach. Various characterization tools were employed to support the experimental findings. Notably, impressive schematic visualizations were prepared to address gaps in characterization, such as the absence of TEM analysis. Additionally, colorful and well-designed figures were included to enhance clarity and engage readers, aligning with the expectations of a high-impact journal. However, relying heavily on schematic designs rather than precise experimental methods can sometimes lead to misinterpretation. Incorporating modeling tools alongside experimental data would significantly enhance the accuracy of the presented findings. Therefore, it is recommended that the manuscript be revised and submitted to a journal that is more aligned with its current scope and quality since it does not meet the standards of this high-impact journal. Here are some comments.

1. As indicated in the title, the synergistic effects should be clearly defined in the introduction and consistently emphasized throughout the manuscript, including the conclusion. Currently, it is unclear what specific synergistic effects are being highlighted. A slight revision of the introduction section could help clarify the focus and better align it with the title, making it more engaging. Given the abundance of studies on electrode materials for enhancing activity, stability, and cost-effectiveness, the concise discussion would make the manuscript stand out.

2. Throughout the manuscript, mechanisms are frequently mentioned, but without supporting theory or modeling, these discussions should be minimized or thoroughly substantiated using references. Specifically, without routine TEM analysis and other characterization approaches of materials, the validity of the proposed hypothesis is questionable. While the method is not novel, it is a good application for energy devices. A similarity check with existing literature should be incorporated into the introduction and the sections on materials synthesis and fabrication to ensure originality.

3. A more detailed discussion should be included on why PNC was chosen over other materials. While Figure 1 provides a solid comparison between this study and previous works, differences in experimental conditions (as noted in the supporting information) must be addressed to avoid misleading readers. Additionally, claims about scalability should be revised, as the current approach does not appear significantly easier to implement compared to traditional methods.

4. While this study reports high-performance PCECs and acknowledges the researchers' efforts, some wording should be avoided unless supported by accurate experimental evidence. For example, on Page 3, Line 89, where is the experimental evidence demonstrating surface activity, mass transfer, and interfacial stability? On Page 4, Lines 137 and 150, the term 'optimal' porosity is used. However, the process or data showing how this 'optimal' result was achieved and its relevance to the overall research are not clearly presented. In addition, on Page 6, Line 175, the term 'complicated relationship' requires further clarification. Although additional examples are not included in this review, they should be carefully identified and addressed to enhance clarity and credibility throughout the manuscript.

5. Including schematic drawings is appreciated; however, they focus only on surface-level representations and lack vital factors. The presented concepts are macroscopic, but after reduction, pathways for O^{2-} via bulk PNC73 should be discussed to better align with high-impact findings. Does it occur only via the surface diffusion to the electrolyte? Incorporating atomic-

level insights would significantly enhance the quality and depth of understanding of the high performance reported in this study. The application of PNC ink does not seem appropriate compared to traditional approaches. In particular, Figure 4 relies heavily on conceptual ideas without sufficient supporting evidence.

6. Despite the high performance for practical applications, a significant limitation of this study is the presentation of nearly identical activation energies in Figure 3f,g, which are approximately 0.11 eV and 0.3 eV. It raises questions about the differentiation and significance of these results. A more detailed and thorough data analysis would be valuable for better interpretation and justifying the reported high performance. Also, the discussion of cost-effectiveness on Page 10, Line 2, should be compared with reference materials to provide a clearer argument.

7. Figure 6b should be revised to clarify its purpose. As they conducted the 100-hour tests, Figures 6d,e should be expanded to emphasize the significance of the results by showing the whole range of the 100-hour tests. Similar issues are present in Figure 7, where additional analysis and emphasis on key findings would enhance its impact.

Reviewer #2

(Remarks to the Author)

Dear Editor,

This article demonstrates the application of a nano-architected ultra-porous oxygen electrode in proton ceramic electrochemical cells (PCECs). The authors highlight that the utilization of this structured electrode results in remarkable electrochemical performance and good durability. Overall, the manuscript is well-organized, presents crystalized data, and draws convincing conclusions. However, several points need to be addressed before the manuscript can be considered for acceptance:

1. Both the PNC and mesh-like air electrodes have been previously reported in the authors' earlier publications (Nature Communications, 2020, 11(1): 1907; Advanced Science, 2018, 5(11): 1800360). The authors should explicitly highlight the novelty and new findings of this work in the introduction part.
2. Figure 5C-G, the authors report very high current densities during steam electrolysis. However, these tests were conducted with the air electrode fed with a water vapor and pure O₂ mixture. P-type air electrodes naturally exhibit higher conductivity and much faster ORR/OER kinetics under elevated oxygen partial pressures. In contrast, the reference data in Figure 5G were obtained under vapor-air mixtures, so that it is inappropriate to directly compare the reported electrochemical performance with the results shown in Figure 5G. The authors should address this inconsistency.
3. Faradaic efficiency is a critical metric for steam electrolysis in PCECs. Without this data, the reported current density values are of limited significance. I strongly recommended that the authors provide Faradaic efficiency results, particularly at high current densities.
4. Figure 3H. The authors observed a decrease in activation energy with the use of the new electrode and attributed this to enhanced mass transport. However, activation energy is fundamentally related to the reaction energy barrier, and providing more reaction sites, in principle, does not alter the activation energy barrier. The authors should clarify and provide a detailed explanation of this observation.
5. On page 14, line 392, the authors observed a quicker recovery rate of reaction activity at higher temperatures. The authors should explain why elevated temperatures enable faster recovery of electrochemical performance.
6. Figure 3D. The authors observed a decrease in bulk resistance under electrolysis mode. They should provide a detailed explanation of this phenomenon and discuss whether this could affect Faradaic efficiency.

Reviewer #3

(Remarks to the Author)

This is a very important work for SOFCs. The authors use cotton fiber as the template for electrodes for reversible SOFCs. Very impressive performance with excellent stability have been achieved. However, more experimental details should be added, allowing other researchers to repeat this work and to use the method for other materials. The paper can be accepted for publication after addressing the following issues.

line 462, 'The prepared cotton fiber ...' How was the key cotton fiber prepared ? The detailed preparation method should be included.

line 463, 'dissolve the stoichiometric amount of PNC73 powder in 95% ethanol and agitate for half an hour until a clear phase is produced' Are you sure that the oxide can be dissolved in ethanol to obtain a clear solution ?

line 466, 'After fully immersing the fiber for 2 hours, it was calcined at 750°C for 3 hours.' Which atmosphere, in air or under the protection of other gases ? What was the heating/cooling rate ?

The key electrolyte material, BaCe_{0.4}Zr_{0.4}Y_{0.1}O_{3-δ} (BZCYYb4411), was first reported by Prof. Meilin Liu, published in Science. That key paper should be cited.

Version 1:

Reviewer comments:

Reviewer #1

(Remarks to the Author)

The manuscript has been thoroughly revised, satisfactorily addressing all of the reviewers' comments and suggestions. Therefore, it is now suitable for acceptance and publication in this journal.

Reviewer #2

(Remarks to the Author)

The authors have addressed my questions. I recommend accepting this manuscript for publication.

Reviewer #3

(Remarks to the Author)

The author has not answered my question regarding the firing atmosphere.

'...it was calcined at 650°C/750°C/850°C for 3 hours with the heating rate of 1°C/minute and the cooling rate of 3°C/minute to get mesh-structure electrodes with varying porosity.'

Was it fired in air ? If yes, can you add 'in air' to clarify this ?

After this correction, it can be accepted for publication and I do not need to view it again.

Point-to-point responses to the reviewers' comments and suggestions

Reviewer # 1

Comments to the Author

Zheng et al. present a study on an enhanced cathode material for protonic ceramic electrochemical cells (PCECs), developed using a nano-fiber-structured ultra-porous design approach. Various characterization tools were employed to support the experimental findings. Notably, impressive schematic visualizations were prepared to address gaps in characterization, such as the absence of TEM analysis. Additionally, colorful and well-designed figures were included to enhance clarity and engage readers, aligning with the expectations of a high-impact journal. However, relying heavily on schematic designs rather than precise experimental methods can sometimes lead to misinterpretation. Incorporating modeling tools alongside experimental data would significantly enhance the accuracy of the presented findings. Therefore, it is recommended that the manuscript be revised and submitted to a journal that is more aligned with its current scope and quality since it does not meet the standards of this high-impact journal. Here are some comments.

Response: We sincerely thank Reviewer #1 for the thoughtful and constructive feedback, which has greatly contributed to enhancing the quality of our manuscript. In response to the comments provided, we have conducted additional experimental studies to address the reviewer's concerns and to provide stronger evidence supporting our hypothesis. Below, we present detailed responses to each comment, along with the corresponding revisions made in the manuscript.

1) As indicated in the title, the synergistic effects should be clearly defined in the introduction and consistently emphasized throughout the manuscript, including the conclusion. Currently, it is unclear what specific synergistic effects are being highlighted. A slight revision of the introduction section could help clarify the focus and better align it with the title, making it more engaging. Given the abundance of studies on electrode materials for enhancing activity, stability, and cost-effectiveness, the concise discussion would make the manuscript stand out.

Response: Thank you for highlighting the key concept of our study - synergy.

To improve our manuscript particularly the discussions about synergistic impact of NAUP electrode, we have expanded related contents in introduction section and discussion section to better elucidate our hypothesis and how it improves PCEC performances and various durability behaviors. As the reviewer raised the next question, which is more specific on synergy, we would like to combine these two comments and provide the replies under Q#2 by providing more substantial evidence to support the proposed synergistic mechanism.

2) *Throughout the manuscript, mechanisms are frequently mentioned, but without supporting theory or modeling, these discussions should be minimized or thoroughly substantiated using references. Specifically, without routine TEM analysis and other characterization approaches of materials, the validity of the proposed hypothesis is questionable. While the method is not novel, it is a good application for energy devices. A similarity check with existing literature should be incorporated into the introduction and the sections on materials synthesis and fabrication to ensure originality.*

Response: Thank you for your positive feedback on the potential applications of this method for energy devices.

(1) Our work demonstrates the improved activity and stability attributed to the proposed nano-fiber structured ultra-porous electrode and underlying synergy can be well supported by our results and more detailed discussion based on this new version.

To clearly explain the synergistic role, below we summarize the critical points to highlight the relationship:

1) At macroscopic level: Our NAUP electrode demonstrates a remarkable synergy by integrating high porosity and nano-scale structural features to optimize both mass transfer and reaction kinetics, which are crucial for efficient hydrogen production. The channel-structured fibers provide high porosity, ensuring adequate mass transfer of steam to supply sufficient reactants for the reaction. Simultaneously, the nano-scale electrode particles accelerate reaction kinetics by offering an enhanced surface area for catalytic activity. This interplay creates a feedback loop: the fast reaction kinetics demand and are supported by efficient mass transfer, while enhanced mass transfer further accelerates the reactions. Additionally, the electrode's design facilitates electron and ion diffusion, seamlessly supporting the reaction process. This synergistic integration of structural and functional properties positions the NAUP electrode as a highly effective solution for achieving rapid and efficient hydrogen production.

2) At microscopic level: the PNC electrode exhibits exceptional synergy as a triple-conducting material, enabling the conduction of protons, oxygen ions, and electrons. This unique conduction behavior plays a pivotal role in facilitating both surface reaction kinetics and bulk diffusion processes. The efficient surface reactions produce protons, which are swiftly transported through the electrode via proton conduction, ensuring uninterrupted reaction progression. Simultaneously, the rapid removal of intermediates, facilitated by oxygen ion and electron conduction, further accelerates the reaction by minimizing the accumulation of reaction by-products. This dynamic interplay between fast surface reaction kinetics and efficient transport of reaction products and intermediates creates a highly synergistic system, driving proton diffusion and oxygen evolution with remarkable efficiency. Such microscopic-level synergy underscores the PNC electrode's potential as a transformative solution for high-performance electrochemical applications.

We have summarized the synergy in the Table 1.

The related contents have been added into manuscript to improve the discussion on describing the synergistic effects.

Synergistic actions	Pros from NAUP (mesh structure)	Evidence
Macroscopic Mass transfer Reaction kinetics Proton conduction	Nano-structured Ultra-porous	SEM  TEM   Microscopic	PNC as a superior TCO material	

Table 1. Correlations between electrochemical actions and NAUP-PNC electrode.

(2) We agree with reviewer that some more characterizations are needed. Therefore, we have carried out more atomic-level characterizations such as TEM analysis, SAED characterization, and detailed comparisons under varied experimental conditions to substantiate our proposed mechanism.

The results of TEM analysis are added to manuscript as Fig. 3.

Fig. 1: TEM characterization of NAUP-PNC oxygen electrode. **A** HRTEM image of NAUP-PNC nanoparticles. The inset contains magnified view of the marked region with a simulated atomic model of the PNC phase (orthorhombic perovskite structure, $Pnma$ space group). **B** SAED pattern of NAUP-PNC nanoparticles. **C** HAADF image of NAUP-PNC nanoparticles. **D** EDS elemental maps of the NAUP-PNC electrode, which confirms the existence of PNC phase. **E** EDS line scan (orange line) of a single NAUP-PNC electrode fiber.

The following results are added to Supplementary Materials as Fig. S4.

Fig. 2: TEM characterization of NAUP-PNC oxygen electrode. **A** HAADF image of NAUP-PNC nanoparticles. **B** Table of elemental amounts in percentages on site 1. **C** EDS spectra of HAADF site 1.

The following paragraphs are added to manuscript from Page 6 Line 159:

“...To investigate the porous nanostructure of the NAUP-PNC electrode, transmission electron microscopy (TEM) was utilized to analyze its microstructure and atomic-level crystal structure. The high-resolution TEM (HRTEM) image of PNC nanoparticles within the mesh fiber, captured near their respective zone axes, is shown in Fig. 3A. The magnified view of the marked region (inset) reveals atomic arrangements consistent with the [111] zone axis of orthorhombic perovskite symmetry. The measured interplanar spacing of 0.339 nm validates the lattice parameter, aligning with the pure PNC phase index information. To further support these observations, a simulated atomic arrangement of the PNC phase is provided, highlighting the atomic pattern. The selected area electron diffraction (SAED) results in Fig. 3B corroborate the specific planes of the PNC crystalline structure. Additionally, a high-angle annular dark-field (HAADF) TEM image, along with elemental distribution maps, is presented in Fig. 3C and Fig. 3D. These maps confirm the uniform distribution of all elements comprising the PNC phase within the selected region (highlighted by the orange rectangle in Fig. 3C). Through the line scan analysis (Fig. 3E) and point analysis (Fig. S3) of a single NAUP fiber in HAADF mode, the elements Pr, Ni, and Co exhibit similar compositional peaks with varying amplitudes along the entire fiber. This consistency indicates that the prepared NAUP mesh retains uniform elemental composition and is well-suited for fabrication onto half cells.”

Details about TEM characterizations are added to Method-Characterizations.

(3) To further confirm the originality of our work, we have conducted a more in-depth similarity analysis with existing literature, particularly focusing on electrochemical performance. We can conclude that our NAUP electrode distinguishes itself from others with superior performance and promising feasibility for industrial applications.

3) A more detailed discussion should be included on why PNC was chosen over other materials. While Figure 1 provides a solid comparison between this study and previous works, differences in experimental conditions (as noted in the supporting information) must be addressed to avoid misleading readers. Additionally, claims about scalability should be revised, as the current approach does not appear significantly easier to implement compared to traditional methods.

Response: We appreciate the reviewer for highlighting this point.

(1) We have added the discussion about why PNC was chosen into the manuscript, and some brief considerations are listed here:

- a) Praseodymium nickel cobaltite (PNC) was chosen as the oxygen electrode material for its exceptional triple-conducting properties and proven durability under extreme operating conditions. As an advanced triple-conducting oxide (TCO), PNC enables the simultaneous transport of protons, oxygen ions, and electrons, setting it apart from conventional mixed ionic and electronic conductors, which typically transport only oxygen ions and electrons. The incorporation of proton conduction into the oxygen electrode significantly enhances the availability of active sites for surface reactions and bulk diffusion. This capability extends the water-splitting reactions across the entire electrode layer, thereby substantially improving the electrochemical performance of the system.
- b) PNC's exceptional hydration capacity significantly increases the proton defect concentration, a key factor for achieving effective triple conduction. Density functional theory (DFT) calculations indicate a substantial reduction in energy barriers, thereby promoting proton conduction¹. Consequently, PNC exhibits outstanding electrochemical performance in both oxygen reduction reactions (ORR) and water oxidation reactions, driven by its high oxygen vacancy concentration, enhanced proton conductivity from hydrated lattices, and low thermal expansion coefficient. These synergistic properties position PNC as a highly promising material for next-generation energy conversion and storage applications.
- c) PNC stands out as an efficient and stable oxygen electrode, eliminating the need for alkaline earth metal doping (e.g., calcium, strontium, or barium) traditionally used to enhance activity. While such doping can improve ionic conductivity and accelerate ORR kinetics, it often leads to interface enrichment and segregation, particularly in environments containing carbon dioxide or water vapor. By avoiding barium doping, PNC overcomes common challenges associated with doped TCOs, such as volatility, lattice strain, and secondary phase formation. This results in improved chemical and structural stability, making PNC a more reliable and durable choice for long-term operation in PCECs.

(2) The performance comparisons have been clarified and more details on operating conditions are added to avoid any misunderstanding:

- a) To address potential concerns regarding the performance comparison in Figure 1, we would like to clarify the rationale behind our comparative analysis, emphasizing the significance of electrode fabrication methods. Our novel electrode employs a mesh-based design, categorizing it as a porous nano-structured electrode. To provide a comprehensive evaluation, we included three widely referenced fabrication types from recent literature: conventional powder-packed electrodes, nano-particle-structured electrodes, and super/ultra-porous electrodes. This approach underscores the unique advantages of the "mesh" design, demonstrating its superior performance and highlighting its potential as a next-generation electrode architecture.
- b) Additionally, we have thoroughly updated and detailed all experimental conditions in Tables S1 and S2 of the Supplementary Materials. These revisions provide readers with a clear understanding of the differences in experimental setups, enabling them to make well-informed assessments of the significance and reliability of our findings.

PPD (W cm ⁻²)	Temperature (°C)								Experimental condition	Fabrication
	700	650	600	550	500	450	400	350		
This work	-	-	1.50	1.17	0.88	0.61	0.41	-	Under H₂/O₂	NAUP
Ref. 15	-	-	0.36	0.22	0.09	-	-	-	A steam partial pressure of 12%/Pure H₂/FC at 0.7 V	3D self-architected steam electrode
Ref. 28	-	-	0.33	0.20	-	-	-	-	Cathode: 75% CO₂+25%O₂/Anode: Ar	3D architected electrode
Ref. 25	-	-	0.92	-	0.58	-	0.21	-	Cathode: pure O₂/Anode: pure H₂	3D engineering electrode
Ref. 17	-	-	0.95	-	-	-	-	-	Wet H₂ (3% steam) on the fuel side and pure oxygen on the oxygen side	Powder-packed electrode
Ref. 29	-	-	0.65	0.50	0.38	0.24	0.14	0.06	Under H₂/air	Powder-packed electrode
Ref. 30	1.30	0.90	0.40	0.20	-	-	-	-	Under H₂/O₂	Powder-packed electrode
Ref. 18	-	-	0.46	-	-	-	-	-	Under H₂/O₂	Powder-packed electrode
Ref. 31	-	-	-	-	-	-	-	-	-	Powder-packed electrode
Ref. 32	0.40	0.28	0.19	0.11	-	-	-	-	Under H₂/air	Powder-packed electrode
Ref. 33	0.34	-	0.21	-	0.12	-	-	-	Humidified H₂/O₂	Powder-packed electrode
Ref. 34	0.24	0.16	0.10	-	-	-	-	-	Humidified hydrogen (~3% H₂O) /O₂	Powder-packed electrode
Ref. 35	-	-	-	-	0.65	0.34	0.18	-	Humidified hydrogen (~3% H₂O) /air	Nano-network electrode
Ref. 36	0.80	-	-	-	-	-	-	-	Under H₂/O₂	Nano-tailoring electrode
Ref. 37	1.37	-	-	-	-	-	-	-	Humidified hydrogen (~6% H₂O) /air	Nanocatalyst electrode
Ref. 38	0.70	-	-	-	-	-	-	-	Humidified hydrogen (~3% H₂O) /air	Infiltrated nano-electrode
Ref. 39	0.49	0.38	0.29	-	-	-	-	-	Humidified hydrogen (~3% H₂O) /air	Infiltrated multiscale electrode
Ref. 40	-	-	0.15	-	-	-	-	-	Humidified hydrogen (~3% H₂O) /air	Nanostructured electrode
Ref. 41	0.72	0.68	-	-	-	-	-	-	Humidified hydrogen (~3% H₂O) /air	Nanostructured electrode

Table S1. Comparison of peak power density (“-” indicates not available).

R_p (Ω cm^2)	Temperature ($^{\circ}\text{C}$)								Experimental condition	Fabricati on
	700	650	600	550	500	450	400	350		
This work	-	-	0.07	0.09	0.13	0.19	0.22	-	Under H_2/O_2 /at OCV	NAUP
Ref. 15	-	-	0.72	1.13	1.86	-	-	-	A steam partial pressure of 12%/Pure H_2/FC at OCV	3D self- architected steam electrode
Ref. 28	-	-	1.55	2.10	2.57	-	-	-	Cathode: 75% $\text{CO}_2+25\%\text{O}_2/\text{Anode}$: Ar/at OCV	3D architected electrode
Ref. 25	-	-	-	-	-	-	0.71	-	Cathode: pure $\text{O}_2/\text{Anode: pure H}_2$ /at OCV	3D engineering electrode
Ref. 17	-	-	0.38	-	-	-	-	-	Wet O_2 (3% steam) at OCV	Powder- packed electrode
Ref. 29	-	-	-	-	-	-	-	-	-	Powder- packed electrode
Ref. 30	-	-	-	-	-	-	-	-	-	Powder- packed electrode
Ref. 18	-	-	0.44	-	-	-	-	-	Under H_2/O_2 /at OCV	Powder- packed electrode
Ref. 31	0.31	-	0.40	-	-	-	-	-	Steam in anode: 0.4 atm	Powder- packed electrode
Ref. 32	0.17	-	-	2.12	-	-	-	-	Under H_2/air /at OCV	Powder- packed electrode
Ref. 33	0.62	-	0.81	-	-	-	-	-	Humidified H_2/O_2 /at OCV	Powder- packed electrode
Ref. 34	0.32	-	2.09	-	-	-	-	-	Humidified hydrogen (~3% $\text{H}_2\text{O})/\text{O}_2/\text{air}$ /at OCV	Powder- packed electrode
Ref. 35	-	-	-	-	0.25	0.42	1.06	-	Humidified hydrogen (~3% $\text{H}_2\text{O})/\text{air}$ /at OCV	Nano- network electrode
Ref. 36	-	-	-	-	-	-	-	-	-	Nano- tailoring electrode
Ref. 37	0.24	-	-	-	-	-	-	-	Humidified hydrogen (~6% $\text{H}_2\text{O})/\text{air}$ /at OCV	Nanocatalyst electrode
Ref. 38	-	-	-	0.67	-	-	-	-	Humidified hydrogen (~3% $\text{H}_2\text{O})/\text{air}$ /at OCV	Infiltrated nano- electrode
Ref. 39	0.06	0.16	0.39	-	-	-	-	-	Humidified hydrogen (~3% $\text{H}_2\text{O})/\text{air}$ /at OCV	Infiltrated multiscale electrode
Ref. 40	-	-	-	-	-	-	-	-	-	Nanostructur ed electrode
Ref. 41	0.99	1.63	-	-	-	-	-	-	Humidified hydrogen (~3% $\text{H}_2\text{O})/\text{air}$ /at OCV	Nanostructur ed electrode

Table S2. Comparison of R_p between this work and other literature results.

(3) We have validated the potential of scaling up this NAUP electrode for large-scale PCEC manufacturing:

Furthermore, to address the scalability of the NAUP electrode, we have successfully fabricated it in larger dimensions, specifically for 5 cm × 5 cm cells (Fig. 3), using the same straightforward approach outlined in this study. This demonstrates the potential of our method for customization and scalability, making it adaptable for various cell sizes and practical applications.

Fig. 3: 5 cm × 5 cm single full cell using the NAUP electrode (this will be presented in our next work).

4) *While this study reports high-performance PCECs and acknowledges the researchers' efforts, some wording should be avoided unless supported by accurate experimental evidence. For example, on Page 3, Line 89, where is the experimental evidence demonstrating surface activity, mass transfer, and interfacial stability? On Page 4, Lines 137 and 150, the term 'optimal' porosity is used. However, the process or data showing how this 'optimal' result was achieved and its relevance to the overall research are not clearly presented. In addition, on Page 6, Line 175, the term 'complicated relationship' requires further clarification. Although additional examples are not included in this review, they should be carefully identified and addressed to enhance clarity and credibility throughout the manuscript.*

Response:

We appreciate the reviewer's suggestion to provide additional experimental evidence to support our proposed mechanism.

Below we respond your comment one by one:

(1) “For example, on Page 3, Line 89, where is the experimental evidence demonstrating surface activity, mass transfer, and interfacial stability?”

Evidence demonstrating the correlation of surface activity, mass transfer, and interfacial stability:

Regarding the statement on Page 3, Line 89, which pertains to boosted surface activity, mass transfer, and interfacial stability, the ultra-porous nano-structured electrode is widely recognized for its enhanced specific surface area. This correlation is significant for understanding the improved reactive activity and stability.

To validate the promoted surface activity, the nanoscale architecture of NAUP electrode is clearly shown in all the SEM and TEM images (see manuscript Fig. 2 and Fig. 3) contained in the manuscript. Nano-structured electrodes enhance surface activity in PCEC by offering a higher surface area, increased active sites, and improved mass and charge transport. Their porous morphology facilitates better gas diffusion and reaction kinetics, reduces overpotentials, and promotes oxygen vacancy formation and proton transport. Additionally, nano-structured designs lower energy barriers for key reactions and provide better thermal and mechanical stability, making them highly efficient for improving ORR and OER performance in PCECs.

The high porosity of intrinsic mesh also guarantees the faster mass transfer, which is necessary for the improved reactive kinetics of ORR and OER. Especially, the crumpled porous morphology and multi-channel cross section of a single mesh fiber are easily to be located at following images (see manuscript Fig. 2D-2F and 5F).

Interfacial stability, a key advantage of the NAUP electrode, is strongly supported by the results from various stability and durability tests (see manuscript Fig. 7). Our conventional long-term stability test and reversible 100-hour test, as well as the newly designed accelerated transient stress tests, collectively reinforce the claim of interfacial stability. Throughout these tests, the current response remained stable, with degradation rates well within acceptable limits. Importantly, no structural delamination was observed, ensuring that the electrode maintained its electrochemical properties.

(2) “On Page 4, Lines 137 and 150, the term 'optimal' porosity is used. However, the process or data showing how this 'optimal' result was achieved and its relevance to the overall research are not clearly presented.”

To demonstrate the "optimal porosity" of the mesh structure used in this study, we have conducted additional experiments specifically designed to investigate the intrinsic differences in electrochemical performance. These experiments underscore the critical importance of selecting and optimizing the appropriate porosity for achieving superior performance. A detailed description of the methodology and results is provided below.

To determine the optimal porosity, we prepared NAUP electrodes by calcining at 650°C, 750°C, and 850°C using the same soaking process and compared their properties. Scanning electron microscopy (SEM) images revealed distinct morphologies influenced by calcination temperature. At 650°C, the mesh exhibited larger pores, whereas calcination at 850°C produced smaller pores within the same specific area. This observation aligns with the general principle that lower calcination temperatures tend to preserve finer structures and smaller particle sizes due to limited diffusion, reduced sintering, and minimized grain growth, thereby maintaining a high surface area—ideal for catalytic applications. Conversely, higher calcination temperatures promote diffusion, sintering, and crystallinity, resulting in larger particle sizes and a reduced surface area.

Balancing calcination temperature is critical to achieving the desired material properties for specific applications. For optimal performance with the NAUP electrode, the mesh structure must have pores of an appropriate size to provide sufficient pathways for dual mass/charge transfer while maintaining enough active PNC nanoparticles to facilitate necessary reactions. Performance data, including the peak power density in fuel cell mode and current density in electrolysis mode, were analyzed for meshes with varying porosities, confirming that the 750°C calcination temperature yielded the optimal mesh structure. The relevant changes have been incorporated into manuscript Fig. 4, and additional statements have been added to the corresponding paragraph to further substantiate our findings.

Balancing the calcination temperature is crucial for achieving the tailored properties. For optimal performance of the NAUP electrode, the mesh structure must feature pores of an appropriate size to ensure efficient dual mass and charge transfer, while also maintaining sufficient active PNC nanoparticles to facilitate the required reactions. Performance data—encompassing peak power density in fuel cell mode and current density in electrolysis mode—were analyzed for meshes with varying porosities. These analyses confirmed that a calcination temperature of 750°C resulted in the optimal mesh structure. The relevant changes have been reflected in manuscript Figure 4, and additional clarifications have been incorporated into the corresponding section to further strengthen our conclusions.

Fig. 4: Illustration of the integrated NAUP PNC electrode and hybrid structure for optimal dual mass/charge transfer and interfacial strength in PCEC. **A** SEM image of cross section of as-fabricated NAUP-PNC/BZCYYb4411/BZCYYb4411-NiO “sandwiched” structure. Despite the use of a mesh-structured electrode, a strong and secure bonding between the electrode layer and electrolyte is still seen, with no evident signs of delamination. **B** Schematic illustration of occurrence of charge and mass dual-transfer through the single NAUP-PNC fiber inner wall to slight loading of PNC glue ink, ultimately to the electrode/electrolyte interface under the fuel cell mode. During the former fabrication, 10 μl PNC73 ink for certain electrode area (0.178 cm^2) was used to achieve refined bonding (Fig. 2a, Step 3). **C** SEM image of single mesh fiber section, verifying the tight bonding between PNC73 particles (α) and NAUP-PNC structure (β). **D** Schematic illustration of enhanced bonding existing between α (microscale) and β (nanoscale). **E** SEM images of mesh calcinated at 650°C, 750°C and 850°C. **F** Bar chart of the comparison on the PPD and CD at 1.30 V at 600°C for different meshes.

For clarification, we will keep the term “optimal' porosity” on Page 4, however, detailed discussion is added to Page 11, Line 309:

“...surface area created by its ultra-porous structure. To explore the relationship between dual-transfer efficiency and pore size in NAUP electrodes, meshes were prepared at calcination temperatures of 650°C, 750°C, and 850°C, denoted as NAUP-650, NAUP-750, and NAUP-850, respectively. This allowed visualization of the impact of calcination temperature on dual-transfer pathways. As shown in the SEM images of temperature-dependent meshes (Fig. 4E), the lower calcination temperature (650°C) preserved finer structures with smaller particle sizes due to reduced sintering and minimized grain growth, resulting in larger pore sizes that enhance gas diffusion. In contrast, higher calcination temperatures reduced pore size due to enhanced sintering and crystallinity, which led to larger particle sizes and decreased surface area, as observed in the NAUP-850 mesh. To balance pore size and active PNC nanoparticle content for optimal reaction facilitation, electrochemical tests were conducted using single full cells fabricated with meshes of the three porosities (Fig. 4F). Compared to the conventional powder-packed PNC electrode, all NAUP-PNC electrodes demonstrated improved performance. Notably, the NAUP-750 achieved a peak power density improvement of up to 66% at 600°C and higher current density for electrolysis at 1.30 V. These results highlight the enhanced performance of the NAUP-750 mesh structure, attributed to its optimal porosity and dual-transfer efficiency.”

For the section “**Methods-Fabrication of NAUP-PNC73 mesh-structure electrode**”, at Line 551, the statement has been revised to be “... After fully immersing the fiber for 2 hours, it was calcined at 650°C/750°C/850°C for 3 hours to get mesh-structure electrodes with varying porosity.”

(3) *“on Page 6, Line 175, the term 'complicated relationship' requires further clarification”*

The term "complicated relationship" on Page 6, Line 175 has been revised to: “...the impedance spectra from full cells are systematically investigated to analyze the relationship between the refined electrode microstructure and improved interfacial activity.” This revision better aligns with our intent to emphasize the synergistic effect of these factors.

5) *Including schematic drawings is appreciated; however, they focus only on surface-level representations and lack vital factors. The presented concepts are macroscopic, but after*

reduction, pathways for O^{2-} via bulk PNC73 should be discussed to better align with high-impact findings. Does it occur only via the surface diffusion to the electrolyte? Incorporating atomic-level insights would significantly enhance the quality and depth of understanding of the high performance reported in this study. The application of PNC ink does not seem appropriate compared to traditional approaches. In particular, Figure 4 relies heavily on conceptual ideas without sufficient supporting evidence.

Response: Thank you for pointing this out.

We would like to discuss with more details about the considerations from the perspectives of the macroscopic & microscopic concepts and thinking in our work:

Our discussion on the synergistic effects enhancing electrochemical performance goes beyond macroscopic observations to include microscopic insights into the improved triple-conducting efficiency. By leveraging the superior TCO material-PNC, we ensure enhanced electrocatalytic activity for both ORR and OER, with reduced overpotentials. Its high stability, oxygen vacancy concentration, and tailored properties guarantee both durability and versatility in dual-mode operations.

The diffusion of O^{2-} does not solely occur on the material surface through processes like adsorption and dissociation; rather, it primarily involves bulk diffusion via oxygen vacancies in PNC73 until it reaches the interface. This mechanism distinguishes MIECs from TCOs. While MIECs conduct both ions (typically oxygen ions) and electrons simultaneously, TCOs facilitate the transport of multiple species, including protons, oxygen ions, and electrons, rendering them exceptionally versatile for energy and chemical conversion technologies^{2,3}. To achieve superior electrochemical performance, faster reaction kinetics are essential for both ORR and OER, necessitating rapid mass transfer and proton conduction.

Specifically:

For ORR, the key kinetic requirements include efficient oxygen adsorption on the electrode surface and high rates of electron transfer to reduce oxygen into intermediate species such as OOH^* , O^* , and OH^* .

For OER, the critical steps involve the initial adsorption of hydroxide or water and the efficient desorption of molecular oxygen to free up catalytic sites.

Higher electrochemical performance in our NAUP-PNC single cell is driven by the interplay of faster reaction kinetics, enhanced mass transfer, and efficient proton conduction. Faster reaction kinetics ensure the rapid conversion of reactants to products, but achieving this requires enhanced mass transfer to supply reactants and remove products effectively, preventing concentration polarization. At the core of this process is enhanced proton conduction, which minimizes internal resistance and supports a high rate of ionic transport across the cell. Our NAUP-PNC electrode meets these requirements with its exceptional design. The PNC material, as a superior TCO, offers the necessary foundation for efficient proton conduction, while its nano-structured, ultra-porous architecture facilitates effective mass transfer. Together, these features create an optimal

environment for faster reaction kinetics, ultimately maximizing the efficiency and durability of PCECs (Fig. 5).

Fig. 5. Schematic for behaviors of PNC as a TCO.

6) *Despite the high performance for practical applications, a significant limitation of this study is the presentation of nearly identical activation energies in Figure 3f,g, which are approximately 0.11 eV and 0.3 eV. It raises questions about the differentiation and significance of these results. A more detailed and thorough data analysis would be valuable for better interpretation and justifying the reported high performance. Also, the discussion of cost-effectiveness on Page 10, Line 2, should be compared with reference materials to provide a clearer argument.*

Response: Thank you for pointing this out. Our results showed that both ohmic/electrode polarization resistances and activation energies have been reduced by using the NAUP electrode.

We would like to provide more details on the related results:

(a) Reduced Activation Energy:

The activation energy (E_a) was reduced with the implementation of the NAUP-PNC electrode, highlighting its enhanced electrochemical performance (Fig. 6). As shown in Fig. 3F, the E_a derived from ohmic resistance was reduced by 7.37%, decreasing from 0.1139 eV to 0.1055 eV. Similarly, Fig. 3G demonstrates that the E_a from polarization resistance experienced a 7.68% reduction, dropping from 0.3294 eV to 0.3041 eV. This reduction in activation energy is attributed

to the synergistic effects provided by the NAUP electrode, which enhances mass transfer, charge transfer, and reaction kinetics. The reduced E_a aligns with the improved interfacial reaction dynamics and overall material conductivity, further substantiating the electrode's superior performance. Notably, the distinction between the two types of activation energy is important: E_a derived from ohmic resistance reflects the material's bulk conduction ability for ions or electrons, while E_a derived from polarization resistance offers insights into the more complex interfacial reaction dynamics, which typically involve multi-step processes such as adsorption, desorption, and chemical transformations.

(b) Reduced Ohmic and Electrode Polarization Resistances:

Both ohmic resistance and electrode polarization resistance were significantly reduced by incorporating the NAUP-PNC electrode, further reinforcing its role in enhancing performance. This improvement is largely due to the electrode's nano-structured and ultra-porous design, which increases the number of active reaction sites and promotes faster mass and charge transfer. The nanostructure not only provides a larger specific surface area for electrochemical reactions but also creates more efficient pathways for ion and electron transport, directly boosting the kinetics of the reaction.

Together, these features contribute to consistent performance improvements. For example, the reduced activation energy (E_a) from ohmic resistance highlights enhanced ionic and electronic conduction, while the decreased E_a from polarization resistance emphasizes improved interfacial reactions. Specifically, the nanostructure alleviates interfacial reaction bottlenecks by facilitating the adsorption, desorption, and chemical transformation processes that govern polarization resistance. As a result, the synergistic effects of the NAUP electrode's material properties ensure reductions in both resistances and activation energy, leading to substantial improvements in electrochemical performance across various operating modes.

Fig. 6: Comparison between E_a from ohmic resistance and E_a from polarization resistance.

(c) We consider the NAUP electrode to be highly promising for PCEC applications, with significant potential for cost-effectiveness. Several factors contribute to this potential: a) the use of a low-cost mesh template made from cloth textile, b) the ease of scaling up production, and c)

the rapid synthesis process. However, since we currently do not have specific cost figures for comparison, we have omitted this information to avoid any confusion.

7) Figure 6b should be revised to clarify its purpose. As they conducted the 100-hour tests, Figures 6d,e should be expanded to emphasize the significance of the results by showing the whole range of the 100-hour tests. Similar issues are present in Figure 7, where additional analysis and emphasis on key findings would enhance its impact.

Response: Thank you for your suggestion.

Regarding the details in Fig. 6B, 6D, and 6E, these represent two separate measurements: a) Fig. 6B shows constant electrolysis testing at a fixed voltage of 1.30V for 50 hours, followed by 1.40V for approximately 55 hours; b) Fig. 6D and 6E were obtained from another set of tests where the cell was dynamically operated across different modes.

To avoid any confusion, we have added additional clarifications in the figure captions for the readers' understanding.

Furthermore, we have expanded the manuscript with relevant discussions on the durability and stability tests of NAUP single full cells, as presented in Fig. 6 and Fig. 7. The following points summarize the findings and their implications:

(a) Long-Term Stability Tests (Fig. 6A and 6B):

- The 100-hour stability test demonstrated minimal degradation under constant applied voltage, with stable current response over time.
- Impedance curves collected 20 times during the test showed nearly overlapping values for ohmic and polarization resistance, confirming consistent performance.
- The alignment of I-V degradation with impedance changes indicated an absence of complex or localized degradation processes, unlike conventional powder-packed electrodes.

(b) Accelerated Transient Tests (Fig. 6D and 6E):

- Dynamic tests under rapidly changing conditions revealed degradation rates below 2.6%, highlighting the superior stability and durability of NAUP full cells.
- These tests simulate realistic operational challenges and validate the electrode's ability to handle variable potentials effectively.

(c) Thermal Cycling Tests (Fig. 7):

- Tests conducted at 600°C, 550°C, and 500°C under start-up, shut-down, and fluctuating temperature conditions confirmed consistent resistance and current-voltage behavior.

- The results validated NAUP cells' reliability under dynamic environments, despite the stability challenges posed by thermal expansion and microstructural changes.

These discussions have been incorporated into the manuscript to provide a comprehensive understanding of the stability and reliability of NAUP full cells under realistic and dynamic operational conditions.

Reviewer # 2

Comments to the Author

This article demonstrates the application of a nano-architecture ultra-porous oxygen electrode in proton ceramic electrochemical cells (PCECs). The authors highlight that the utilization of this structured electrode results in remarkable electrochemical performance and good durability. Overall, the manuscript is well-organized, presents crystalized data, and draws convincing conclusions. However, several points need to be addressed before the manuscript can be considered for acceptance:

Response: We sincerely thank Reviewer #2 for their valuable feedback, which has greatly enhanced the quality of our manuscript. To address the reviewer's comments, we conducted additional electrochemical tests, including Faradaic efficiency measurements, to further support our hypothesis on the observed synergy. Below, we provide detailed responses to each comment.

1) Both the PNC and mesh-like air electrodes have been previously reported in the authors' earlier publications (Nature Communications, 2020, 11(1): 1907; Advanced Science, 2018, 5(11): 1800360). The authors should explicitly highlight the novelty and new findings of this work in the introduction part.

Response: We appreciate the Reviewer 2's insightful comment regarding the need to explicitly highlight the novelty and new findings of this work. In response, we have revised the introduction section to emphasize the following innovations and contributions regarding our synergistic mechanism that distinguish this study from our earlier publications:

(a) Innovative Electrode Design:

- Introduced a "homogeneous composition but heterogeneous morphology" configuration in NAUP-PNC full cells, ensuring strong physical and chemical stability despite differing morphologies.
- Utilized PNC ink as an adhesive agent to couple large-grain electrolytes with nanostructured NAUP-PNC electrodes, creating tight bonding interfaces.

(b) Enhanced Mass and Charge Transfer:

- The mesh-structured NAUP-PNC electrode facilitates massive gas transport and efficient catalysis through interconnected gas channels and nanoparticles.
- The hybrid electrode's ultra-porous structure significantly increases the specific surface area, offsetting potential losses in transfer efficiency from reduced thickness (~10 μm).

(c) Dual-Transfer Pathway Optimization:

- Enabled oxygen ions to achieve bulk diffusion and incorporation along enlarged dual-transfer pathways at the microscale and nanoscale interfaces.
- Identified "bridge-shaped" NAUP-PNC structures as critical for enhancing mass transport and charge transfer efficiency.

(d) Pore Size and Calcination Temperature Tuning:

- Demonstrated the effect of calcination temperature (650°C, 750°C, 850°C) on pore size, sintering, and nanoparticle content, balancing gas diffusion and active surface area.
- NAUP-750 mesh was found to provide the best balance, delivering up to 66% higher peak power density at 600°C compared to conventional powder-packed electrodes.

(e) Synergistic Performance Improvements:

- The NAUP-750 mesh-structured electrode demonstrated reduced activation energy for ORR/OER and overall resistance (particularly polarization resistance, R_p).
- Achieved superior electrochemical performance with enhanced current density and cost-effectiveness, making it a promising innovation for energy applications.

Moreover, more significances regarding our highly improved performance using NAUP electrode are also listed:

(a) Exceptional Durability Under Harsh Conditions:

- Demonstrated stable performance during electrolysis at 600°C, with current density decreases of only 1.03% and 1.24% over extended operation at 1.30 V and 1.40 V, respectively.
- Polarization resistance and ohmic resistance remained steady with minimal fluctuations (~5%).

(b) Reversible Operation Capability:

- Achieved seamless transitions between fuel cell and steam electrolysis modes with degradation rates of 1.87% (electrolysis) and 1.55% (fuel cell mode), showcasing flexibility and adaptability for reversible operations.

(c) Resilience Under Escalated Transient Testing:

- Withstood step-voltage transitions (1.50 V, 1.35 V, 1.20 V) with degradation rates below 2% and minimal structural damage, maintaining reliable current density responses.
- Interval-based voltage variations between 1.20 V and 1.50 V showed clear periodic patterns with minimal deviations and a statistical degradation rate of only 2.59%.

(d) Industrial-Scale Potential:

- The robust performance under high overpotential and dynamic conditions highlights the NAUP-PNC full cells' suitability for industrial-scale applications requiring stability and durability.

These innovations highlight the transformative design and optimization of NAUP-PNC electrodes for improving interfacial properties and achieving higher performance.

2) Figure 5C-G, the authors report very high current densities during steam electrolysis. However, these tests were conducted with the air electrode fed with a water vapor and pure O₂ mixture. P-type air electrodes naturally exhibit higher conductivity and much faster ORR/OER kinetics under elevated oxygen partial pressures. In contrast, the reference data in Figure 5G were obtained under vapor-air mixtures, so that it is inappropriate to directly compare the reported electrochemical performance with the results shown in Figure 5G. The authors should address this inconsistency.

Response: Thank you for pointing this out and we have corrected it according to your suggestion.

We agree with reviewer that the p-type air electrodes exhibit higher conductivity and faster ORR/OER kinetics under higher oxygen partial pressures. This was also the exact reason why we designed tests under varying oxygen partial pressures and flow rates to explore the dependences on these factors. In our study (as shown in Fig. 5E and 5F and summarized in Table 3), the variations of humidity and oxygen flow rate can impact electrolysis performance. We identified the optimal gas condition: 20 vol.% H₂O with carrier gas of 40 sccm oxygen.

Humidity (vol. % H₂O)	3	10	20	30
CD at 1.30 V (A cm⁻²)	0.873	1.051	1.345	1.285
O₂ flow rate (sccm)	10	20	40	60
CD at 1.30 V (A cm⁻²)	1.205	1.218	1.260	1.229

Table 3. Current density attained at 1.30 V for electrolysis under different O₂ humidity and flow rate.

Since it is a common practice to use oxygen gas for better performances, we also used pure oxygen. As suggested by reviewer, we have replaced the data points in Fig. 5G with the results from other researchers who obtained performance under pure oxygen to ensure consistency and fairness in comparison.

In addition, the reference data used for comparison have been compiled along with their corresponding experimental conditions, as detailed in Table 4 (Supplementary Materials Fig. SX).

Updated Fig. 5G:

PPD (W cm^{-2})				
Ref.	R17 PNC73	R14 BCFZY	R53 PBCC95	This work
Conditions	H ₂ -Humidified O ₂ (50% H ₂ O)	Humidified H ₂ (3% H ₂ O) -Humidified O ₂ (3% H ₂ O)	Humidified H ₂ (3% H ₂ O) -O ₂	H ₂ -Humidified O ₂ (3% H ₂ O)
PPD (W cm^{-2})	0.95	0.88	0.54	1.50

CD at 1.30V (A cm^{-2})				
Ref.	R16 PB30	R53 PBCC95	R54 PNC55	This work
Conditions	H ₂ - Humidified O ₂ (60% H ₂ O)	Humidified H ₂ (3% H ₂ O) -Humidified O ₂ - (20% H ₂ O)	Humidified H ₂ (3% H ₂ O) -Humidified O ₂ (50% H ₂ O)	H ₂ - Humidified O ₂ (20% H ₂ O)
CD at 1.30V (A cm^{-2})	0.83	0.71	0.75	1.35

Table 4. Comparison details for Fig. 5G (manuscript Fig. 6G).

3) *Faradaic efficiency is a critical metric for steam electrolysis in PCECs. Without this data, the reported current density values are of limited significance. I strongly recommended that the authors provide Faradaic efficiency results, particularly at high current densities.*

Response: Thank you for your recommendation.

We agree that the Faradaic efficiency is an important factor to determine the energy efficiency. Therefore, we have carried out the measurements to understand the electronic leakage behavior under different current density. As can be seen in the figure below, FE is >90% at 1.0 A cm^{-2} at 600°C with 20% H₂O, which is consistent with our recent PCEC results. It is believed that electronic leakage is dependent on applied voltage, temperature, and gas condition. We have included this result into Supplementary Materials as Fig. S8.

Fig. 7. Faradic efficiency (%) and Energy efficiency (%) of NAUP-PNC full cells in electrolysis mode at 600°C.

The corresponding discussion was added to Page 13, Line 397.

“... Faradaic efficiency (FE) and energy efficiency (EE) were evaluated for NAUP-PNC full cells to demonstrate their high performance in energy-intensive processes, such as water splitting. During steam electrolysis under 20 vol.% H₂O at 600°C, the cells achieved an impressive FE exceeding 95% at a current density of 0.6 A cm⁻². Additionally, the energy efficiency for converting electrical energy into chemical energy surpassed 80% under the same conditions. These results highlight the exceptional feasibility of hydrogen production using the NAUP-PNC oxygen electrode.”

The measurement procedures have been added to the **Methods** part, titled as “**Faradaic efficiency and Energy efficiency measurements**”.

Faradaic efficiency and Energy efficiency measurements

To evaluate the electrolysis performance for hydrogen production, humidified oxygen gas was supplied to the NAUP electrode with a steam concentration of 20%. Pure hydrogen gas (40 sccm) was fed to the hydrogen electrode as the sweep gas. To evaluate the FE (%), the hydrogen electrode outlet gas was injected into a specially designed flow meter to quantify the hydrogen production rate. The FE (%) was calculated using

$$v = \frac{V_m}{2F} \times i$$

$$FE(\%) = \frac{v_e}{v} \times 100\%$$

where v is the theoretical hydrogen production flow rate (standard cubic centimeter per minute, sccm), i is the input current (A), 2 is the number of electrons involved in the steam electrolysis

reaction, F is the Faraday constant ($9.6485 \times 10^4 \text{ C mol}^{-1}$), and V_m is the molar volume of a gas ($22\,400 \text{ mL mol}^{-1}$). The experimental hydrogen production flow rate v_e .

The electrolysis energy efficiency was calculated according to

$$EE(\%) = \frac{FE(\%)}{v}$$

v is the corresponding voltage (V).

4) Figure 3H. The authors observed a decrease in activation energy with the use of the new electrode and attributed this to enhanced mass transport. However, activation energy is fundamentally related to the reaction energy barrier, and providing more reaction sites, in principle, does not alter the activation energy barrier. The authors should clarify and provide a detailed explanation of this observation.

Response: Thank you for your comment regarding the relationship between decreased activation energy and enhanced mass transport.

We agree that intrinsic activation energy is intrinsically linked to the reaction energy barrier and that, in principle, increasing the number of reaction sites does not directly impact the activation energy barrier.

In our work, we calculated from the dependence of polarization resistance over temperature to give the apparent activation energy, which can be affected by decreasing catalyst size to the nanoscale:

(1) Nanometer-sized catalysts have a much higher surface area-to-volume ratio, which increases the number of active sites available for reaction. This enhanced interaction between the reactant molecules and the catalyst surface can lead to a reduction in the apparent activation energy because the reaction becomes more efficient. In practice, this often appears as a reduced energy barrier when analyzed macroscopically because the system is utilizing the active sites more efficiently⁴.

(2) Quantum effects and altered surface energies: As the TEM revealed, the size of nanoparticles is about tens of nanometers. At the nanoscale, catalysts often exhibit quantum effects and altered surface electronic properties. These changes can stabilize transition states or reaction intermediates, effectively lowering the energy barrier for the reaction.

(3) Enhanced reactant adsorption and diffusion: Nanosized catalysts can promote better adsorption of reactants and facilitate faster diffusion to active sites. These improvements can increase the reaction rate, making it appear as though the activation energy is reduced. Our work focuses on optimizing porosity and pore connectivity in the NAUP electrode to enhance mass transfer pathways and minimize diffusion resistance. This is achieved through the design of hierarchical structures with micro- and nano-scale features, leveraging the synergistic effects of the NAUP-PNC electrode and PNC ink. Enhanced diffusion pathways ensure that the transport of reactants (e.g., oxygen or water vapor) through the porous electrode structure matches the reaction rate determined by the activation energy barrier.

Within this optimized framework, we observed a slight but consistent decrease in activation energy values: a 7.37% reduction for E_a from ohmic resistance and a 7.68% reduction for E_a from polarization resistance. The rate-determining steps for ORR and OER are closely associated with the adsorption/desorption efficiency of intermediates. The ultra-porosity and expanded active sites of the NAUP electrode stabilize these intermediates, facilitating various reaction steps.

We have included some discussion into the manuscript.

5) On page 14, line 392, the authors observed a quicker recovery rate of reaction activity at higher temperatures. The authors should explain why elevated temperatures enable faster recovery of electrochemical performance.

Response: Thank you for your comment.

The faster recovery of PCEC performance at higher temperatures can be attributed to thermally activated processes that enhance ionic and protonic transport, defect equilibration, surface catalysis, stress relief, and structural recovery. These mechanisms collectively restore the system to its optimal state.

Firstly, at higher temperatures, proton and oxygen ion transport within the electrolyte and electrodes becomes thermally activated⁵. This increased mobility facilitates the rapid restoration of charge carrier balance. Proton conduction relies on hopping mechanisms between lattice sites or along grain boundaries, which operate more efficiently at elevated temperatures due to reduced energy barriers.

Secondly, higher temperatures enhance the diffusion of oxygen vacancies, allowing the system to re-establish its optimal defect configuration⁶. This also facilitates the re-equilibrium of structural and chemical defects that may have accumulated during operation or cycling, further contributing to performance recovery.

Given these factors, the observed degradation rates of current densities—1.12% during steam electrolysis and 1.76% during fuel cell operation at 600°C—are understandable and consistent with the beneficial effects of thermal activation on performance recovery.

6) Figure 3D. The authors observed a decrease in bulk resistance under electrolysis mode. They should provide a detailed explanation of this phenomenon and discuss whether this could affect Faradaic efficiency.

Response: Thank you for your comment.

The reviewer raised a very interesting topic for discussion. Yes, it is very common to observe the decrease of ohmic resistance under electrolysis mode when running impedance spectra. Here are some examples from other researchers:

- a) Babiniec et al. found that values of electrode polarization resistances gradually decrease by 33% with the applied voltage and attributed it to the growing electronic conductivity in electrolytes⁷.

- b) Gan et al. showed the maximum value of resistance gradually declines in the degree of around 40% with voltage increasing from 0 to 2 V, which may be attributed to the contribution of p-type conduction caused by the oxidizing conditions on the oxygen electrode side⁸.
- c) Li and Xie revealed that ohmic resistance decreases slightly while the polarization resistance drops significantly with the increase of applied voltage, which may be due to that the reoxidation of electrode enhances the overall conductivity⁹.
- d) Huan et al. examined the dependence of real polarization resistances on electrolysis voltages and water vapor pressure¹⁰. It suggested that, in addition to electronic conduction in electrolytes, different electrode reaction steps contribute to the significantly reduced polarization resistance observed in electrolysis mode compared to open circuit voltages.

The fundamental mechanisms responsible for this behavior likely involve several factors: (1) Enhanced proton conduction may be due to electrochemical proton injection or increased ion mobility under the influence of an applied electric field. (2) Joule heating or localized hotspots generated during electrolysis can also reduce resistance by raising the electrolyte's temperature.

Regarding the compromised Faradaic efficiency, we acknowledge that it is indeed affected during electrolysis, which is an intrinsic electronic leakage behavior once the voltage is applied, as discussed in response to your question #3. However, we can mitigate electronic leakage by reducing the overpotential on the electrolyte, increasing steam concentration, and operating at lower temperatures. This is a complex issue that we aim to explore further within the PCEC community.

This work holds promise for improving Faradaic efficiency by optimizing electrode polarization, with the potential to achieve efficiencies greater than 90% at 1.0 A cm⁻². We will continue investigating strategies to enhance Faradaic efficiency, as it remains an important focus for future research.

We have added some discussion into the manuscript.

Reviewer #3 (Remarks to the Author):

This is a very important work for SOFCs. The authors use cotton fiber as the template for electrodes for reversible SOFCs. Very impressive performance with excellent stability have been achieved. However, more experimental details should be added, allowing other researchers to repeat this work and to use the method for other materials. The paper can be accepted for publication after addressing the following issues.

Response: We sincerely thank Reviewer #3 for your very positive comments. We have provided more detailed information and explanation to address your concerns.

1) line 462, 'The prepared cotton fiber ...' How was the key cotton fiber prepared? The detailed preparation method should be included.

Response: Thank you for your comment on this.

The expression “The prepared cotton fiber...” was intended to convey that the cotton fiber used in this study is readily available and sourced from commercial markets (FabricLA). To facilitate the soaking process in beakers containing PNC73 precursor solutions of varying concentrations, the cotton fiber was cut into appropriately sized pieces. This information has now been included in the Methods section for clarity.

“The prepared cotton fiber (FabricLA) was cut into 5 x 5 cm² squares and soaked in the PNC73 precursor solution.”

2) line 463, 'dissolve the stoichiometric amount of PNC73 powder in 95% ethanol and agitate for half an hour until a clear phase is produced' Are you sure that the oxide can be dissolved in ethanol to obtain a clear solution?

Response: Thank you for pointing this out.

To address this concern, we have provided additional details on the synthesis of pure PNC73 powder. The sol-gel process was employed to synthesize PNC73 powder for use as the oxygen electrode, with citric acid and EDTA serving as complexing agents.

Specifically, stoichiometric amounts of Pr(NO₃)₃·6H₂O (99.9%, Alfa Aesar), Ni(NO₃)₂·6H₂O (99.9985%, Alfa Aesar), and Co(NO₃)₂·6H₂O (98+%, Alfa Aesar) were dissolved in deionized water along with EDTA and citric acid to prepare a transparent precursor solution. The chelation process was carried out with a molar ratio of EDTA: citric acid: cations = 1:1.5:1, and a cation concentration of 0.02 mol L⁻¹ for this batch. The precursor solution was then magnetically stirred and heated until a viscous gel formed. The gel underwent further heating at approximately 200°C, leading to self-ignition and the formation of powdery ash. This ash was transferred to a muffle furnace for calcination at 1000°C for 5 hours, resulting in a crystalline perovskite phase of PNC73.

Additionally, we acknowledge an error in the expression “dissolve the stoichiometric amount of PNC73 powder in 95% ethanol.” This has been revised to: *“Dissolve the stoichiometric amount of PNC73 powder in the desired solvent (a 1:1 mixture of 95% ethanol and deionized water) and agitate for half an hour until a clear phase is obtained.”*

3) line 466, 'After fully immersing the fiber for 2 hours, it was calcined at 750°C for 3 hours.' Which atmosphere, in air or under the protection of other gases? What was the heating/cooling rate?

Response: Thank you for comment on these details we should provide.

Line 466, it has been revised to be *with the detail that “with the heating rate of 1°C/minute and the cooling rate of 3°C/minute.”*

4) The key electrolyte material, BaCe_{0.4}Zr_{0.4}Y_{0.1}Yb_{0.1}O_{3-δ} (BZCYYb4411), was first reported by Prof. Meilin Liu, published in Science. That key paper should be cited.

Response: Thank you for pointing this out.

We have cited the paper you mentioned. More details can be located in Reference part.

References

1. Ding, H. *et al.* Self-sustainable protonic ceramic electrochemical cells using a triple conducting electrode for hydrogen and power production. *Nat Commun* **11**, 1907 (2020).
2. Ren, R. *et al.* Tuning the defects of the triple conducting oxide $\text{BaCo}_{0.4}\text{Fe}_{0.4}\text{Zr}_{0.1}\text{Y}_{0.1}\text{O}_{3-\delta}$ perovskite toward enhanced cathode activity of protonic ceramic fuel cells. *J. Mater. Chem. A* **7**, 18365–18372 (2019).
3. Park, K. *et al.* Water-mediated exsolution of nanoparticles in alkali metal-doped perovskite structured triple-conducting oxygen electrocatalysts for reversible cells. *Energy Environ. Sci.* **17**, 1175–1188 (2024).
4. Mao, Z. & Campbell, C. T. Apparent Activation Energies in Complex Reaction Mechanisms: A Simple Relationship via Degrees of Rate Control. *ACS Catal.* **9**, 9465–9473 (2019).
5. Patil, T. C., Duttagupta, S. P., Kulkarni, S. G. & Phatak, G. J. Oxygen ion transport through the electrolyte in Solid Oxide Fuel Cell. in *2013 International Conference on Renewable Energy Research and Applications (ICRERA)* 70–72 (IEEE, Madrid, Spain, 2013). doi:10.1109/ICRERA.2013.6749728.
6. Gunkel, F., Christensen, D. V., Chen, Y. Z. & Pryds, N. Oxygen vacancies: The (in)visible friend of oxide electronics. *Applied Physics Letters* **116**, 120505 (2020).
7. Babiniec, S. M., Ricote, S. & Sullivan, N. P. Characterization of ionic transport through $\text{BaCe}_{0.2}\text{Zr}_{0.7}\text{Y}_{0.1}\text{O}_{3-\delta}$ membranes in galvanic and electrolytic operation. *International Journal of Hydrogen Energy* **40**, 9278–9286 (2015).
8. Gan, Y. *et al.* Composite Oxygen Electrode Based on LSCM for Steam Electrolysis in a Proton Conducting Solid Oxide Electrolyzer. *J. Electrochem. Soc.* **159**, F763–F767 (2012).
9. Li, S. & Xie, K. Composite Oxygen Electrode Based on LSCF and BSCF for Steam Electrolysis in a Proton-Conducting Solid Oxide Electrolyzer. *J. Electrochem. Soc.* **160**, F224–F233 (2013).
10. Huan, D. *et al.* Investigation of real polarization resistance for electrode performance in proton-conducting electrolysis cells. *J. Mater. Chem. A* **6**, 18508–18517 (2018).

Point-to-point responses to the reviewers' comments and suggestions

Reviewer #1 (Remarks to the Author):

The manuscript has been thoroughly revised, satisfactorily addressing all of the reviewers' comments and suggestions. Therefore, it is now suitable for acceptance and publication in this journal.

Response: We sincerely thank you for your feedback. I appreciate the reviewers' comments and am glad to hear that the revisions have met the necessary standards.

Reviewer #2 (Remarks to the Author):

The authors have addressed my questions. I recommend accepting this manuscript for publication.

Response: Thank you for your recommendation. I'm pleased to hear that your questions have been satisfactorily addressed. I appreciate your support for the manuscript's acceptance for publication.

Reviewer #3 (Remarks to the Author):

The author has not answered my question regarding the firing atmosphere.

'...it was calcined at 650°C/750°C/850°C for 3 hours with the heating rate of 1°C/minute and the cooling rate of 3°C/minute to get mesh-structure electrodes with varying porosity.'

Was it fired in air ? If yes, can you add 'in air' to clarify this ?

After this correction, it can be accepted for publication and I do not need to view it again.

Response: We sincerely thank you for your valuable comment. Yes, it was calcined in ambient air and I have added "in ambient air" to clarify the firing atmosphere in the manuscript.